# WeatherGFM: Learning A Weather Generalist Foundation Model via In-context Learning

**Xiangyu Zhao[2, 1,†] Zhiwang Zhou[1], Wenlong Zhang[1,✉], Yihao Liu[1], Xiangyu Chen[1],**
**Junchao Gong[1, 3], Hao Chen[1], Ben Fei[1], Shiqi Chen[4], Wanli Ouyang[1],**
**Xiao-Ming Wu[2, ✉], Lei Bai[1]**
[1]Shanghai AI Laboratory   [2]The Hong Kong Polytechnic University
[3]Shanghai Jiao Tong University   [4]Shanghai Meteorological Service
zhangwenlong@pjlab.org.cn, xiao-ming.wu@polyu.edu.hk

## Abstract

The Earth's weather system involves intricate weather data modalities and diverse weather understanding tasks, which hold significant value to human life. Existing data-driven models focus on single weather understanding tasks (e.g., weather forecasting). While these models have achieved promising results, they fail to tackle various complex tasks within a single and unified model. Moreover, the paradigm that relies on limited real observations for a single scenario hinders the model's performance upper bound. Inspired by the in-context learning paradigm from visual foundation models and large language models, in this paper, we introduce the first weather generalist foundation model (WeatherGFM) to address weather understanding tasks in a unified manner. Specifically, we first unify the representation and definition for diverse weather understanding tasks. Subsequently, we design weather prompt formats to handle different weather data modalities, including single, multiple, and temporal modalities. Finally, we adopt a visual prompting question-answering paradigm for the training of unified weather understanding tasks. Extensive experiments indicate that our WeatherGFM can effectively handle up to 12 weather understanding tasks, including weather forecasting, super-resolution, weather image translation, and post-processing. Our method also showcases generalization ability on unseen tasks. The source code is available at https://github.com/xiangyu-mm/WeatherGFM.

## 1 Introduction

Modeling Earth weather systems involves a series of complex subprocesses that are intended to transform intricate Earth observation data into applications like weather forecasting (Chen et al., 2023a; Bi et al., 2023), downscaling (Chen et al., 2022), assimilation (Huang et al., 2024), retrieval (Liu et al., 2011), and bias correction (Gong et al., 2024). During the past decade, many data-driven machine learning methods have been investigated for various weather understanding tasks and delivering desirable performance on specific tasks. For example, recent studies using large-scale training data (e.g., ERA5 reanalysis data (Hersbach et al., 2020)) have exceeded the accuracy of conventional numerical weather forecasts. However, current weather foundational models face challenges regarding generalizability and data scale limitations. On the one hand, the Earth observation system consists of a variety of observation devices, such as satellites, radar, and weather stations, which produce diverse modalities of data. Consequently, designing a specific model for a single-task scenario is highly complex, time-consuming, and labor-intensive. On the other hand, large-scale data in fields such as computer vision can be obtained at a low cost, whereas weather understanding tasks face an intrinsic bottleneck in data scale due to restrictions on individual scenes and single observation devices as shown in Table 1. For instance, local short-term precipitation forecasting models can only utilize a finite range of observational data.

---

†This work was primarily conducted during the author's internship at the Shanghai Artificial Intelligence Laboratory.

A significant trend in AI research is the development of foundation models, shifting towards large-scale pre-training and in-context learning. This paradigm enables unified processing of a multitude of complex tasks and generalization to unseen tasks. For example, large language models (LLMs) can perform a variety of language-centric tasks (e.g., sentiment analysis, question answering and machine translation) by combining language input-output examples with new query inputs (prompts) without optimizing model parameters (Brown, 2020). Similarly, vision foundation models (Wang et al., 2023b; Liu et al., 2023b; Chen et al., 2024b) employ visual prompts with query inputs to carry out diverse image-centric tasks, such as semantic segmentation, depth estimation, and image restoration. These studies highlight the significant potential of generalist foundational models.

The study of foundation models remains largely limited in weather understanding, with the majority focused on Computer Vision and Natural Language Processing. While there has been some progress with large foundation models in weather and climate, the focus is mainly on weather forecasting and downscaling tasks. For example, Climax (Nguyen et al., 2023) uses a pre-training-finetuning paradigm for weather forecasting and downscaling. Aurora (Bodnar et al., 2024) employs LoRA to unify weather forecasting and quick prediction of atmospheric chemistry. However, as shown in Table 1, these studies do not take into account the modeling of multi-modalities and multi-tasks. This poses a challenge: *Is it possible to design a universal foundation model capable of handling a variety of complex weather understanding tasks and data modalities?*

In this paper, we first propose a weather generalist foundation model, WeatherGFM, to uniformly address a variety of complex weather understanding tasks and data modalities. Unlike prior studies that focused on weather forecasting, our proposed method can expand the task scope to weather forecasting, weather super-resolution (i.e., weather downscaling) (Veillette et al., 2020), weather image translation (similar to retrieval in weather) (Veillette et al., 2020), and post-processing (Gong et al., 2024). These tasks all belong to the domain of weather understanding, but their modalities are distinct. Specifically, Sequence modal data can be utilized for weather forecasting, such as short-term predictions based on radar data. Multi-modal data can be employed for weather image translation, such as converting multi-modal satellite data to generate radar data. Single-modal data can be applied to various common scenarios, such as radar image super-resolution and post-processing. To unify the diverse weather data modalities into a general representation, we introduce a weather prompt format that assigns different prompt phrases to various modalities. By leveraging in-context learning, our WeatherGFM achieves a promising in-context ability on both various seen tasks and unseen tasks. The significance of our work can be summarized as:

- We propose the first weather generalist foundation model (i.e., WeatherGFM), which can handle more than 12 weather understanding tasks.

- Our weather prompt design supports a diversity of weather data modalities, including time-series, multi-modal, and single-modal data.

- Our WeatherGFM with in-context learning first demonstrates the generalization ability to unseen weather understanding tasks.

## 2 RELATED WORK

**Weather understanding and beyond.** Over the past decade, machine learning techniques (Zhang et al., 2022; 2023; 2024) have consistently attracted attention in the field of weather and climate. Numerous data-driven machine learning models have been proposed to address classical tasks in weather understanding (Veillette et al., 2020), such as forecasting, super-resolution, image translation, and post-processing. Weather forecasting (Bi et al., 2023)) aims to predict future observations from past data. Weather super-resolution tasks, i.e., weather downscaling Chen et al. (2022) focus on recovering high-resolution data from low-resolution observations. Weather image translation tasks (Stock et al., 2024) involves converting existing observational data into desired target modalities, such as transforming satellite observations into ground-based weather radar data. Post-processing tasks seek to enhance existing model results, such as bias correction and deblurring (Gong et al., 2024). Despite significant advancements, current methods often rely on specialized datasets and customized single-task models for certain scenarios. Consequently, single-task models struggle to exhibit strong generalization abilities and fail to capture the interconnections between diverse tasks, which hinders the establishment of simulations for the Earth system.

Table 1: Comparison of task-specific models and general models across Earth science and computer vision domains. Our proposed WeatherGFM stands out with its capability to handle multiple tasks, process multi-modal data, and demonstrate general adaptability—underscoring its strength in acquiring challenging-to-access weather data.

| Category | Method | Data Acquisition Difficulty | Supported Tasks | Multi-tasks support? | Multi-modal support? | Generalist support? |
|---|---|---|---|---|---|---|
| Computer Vision | HAT (Chen et al., 2023b) | Low-cost | Image super-resolution (SR) | ✗ | ✗ | ✗ |
| | IPT (Chen et al., 2021) | Low-cost | Image restoration, Derain, Dehaze | ✓ | ✗ | Requires fine-tuning |
| | Painter (Wang et al., 2023a) | Low-cost | Image restoration Segmentation, Keypoint detection | ✓ | ✗ | ✓ |
| | PromptGIP (Liu et al., 2023a) | Low-cost | Image restoration, Derain, Dehaze | ✓ | ✗ | ✓ |
| | GenLV (Chen et al., 2024a) | Low-cost | Image restoration, enhancement, translation | ✓ | ✗ | ✓ |
| Earth Science | Prediff (Gao et al., 2024) | High-cost | Weather forecasting | ✗ | ✗ | ✗ |
| | Cascast (Gong et al., 2024) | High-cost | Post-processing | ✗ | ✗ | ✗ |
| | Climax (Nguyen et al., 2023) | High-cost | Weather forecasting, Super-resolution | ✓ | ✗ | Requires fine-tuning |
| | Aurora (Bodnar et al., 2024) | High-cost | Weather forecasting Atmospheric chemistry prediction | ✗ | ✓ | Requires fine-tuning |
| | WeatherGFM (ours) | High-cost | Weather forecasting, Weather image SR Weather image translation, Post-processing | ✓ | ✓ | ✓ |

**Weather foundation model.** The rise of foundation models (Liu et al., 2024; Zhao et al., 2024a;b; Xu et al., 2023) in Natural Language Processing and computer vision has sparked interest in their application for weather and climate. Large foundation models, enhanced through pre-training, improve the generalization of AI climate models and can be fine-tuned for specific tasks. Pathak et al. (2022) proposed FourCastNet, a climate pre-trained model using Vision Transformer for high-resolution predictions and rapid inference through self-supervised pre-training and autoregressive fine-tuning. Pangu-Weather (Bi et al., 2023) utilizes a 3D Earth-specific Transformer for accurate global predictions. ClimaX (Nguyen et al., 2023) introduces supervised pre-training to weather prediction, offering flexibility for diverse forecasting tasks. A pre-training foundation model usually requires mask modeling for pre-training and then undergoes fine-tuning on specific tasks, such as fine-tuning the pre-trained model on weather forecasting, remote sensing classification and segmentation tasks Bodnar et al. (2024); Cong et al. (2022); Noman et al. (2024); Li et al. (2024).

**Visual in-context learning.** In recent advancements, visual in-context learning has emerged as a promising research area, inspired by the success of language models like GPT-3 (Brown, 2020). These models adapt to various NLP tasks using prompts or in-context examples without extensive retraining. Similarly, in the vision domain, models such as MAE-VQGAN (Hojel et al., 2024) and Painter (Wang et al., 2023b) have begun exploring in-context learning. However, challenges persist, especially in low-level tasks requiring detailed pixel manipulation. To address this, PromptGI (Liu et al., 2023b) and GenLV have incorporated in-context learning concepts into their designs to unify low-level vision tasks with diverse input and output modalities, aiming to develop generalist models. Vision-language models like Unified-IO (Lu et al., 2022) and Unified-IO 2 (Lu et al., 2024) have made significant progress in integrating multiple tasks, highlighting the potential for unified approaches across modalities. Additionally, compositional visual reasoning, exemplified by Visual Programming (Gupta & Kembhavi, 2023), aligns with in-context learning goals by emphasizing visual task synthesis. ViperGPT (Surís et al., 2023) further demonstrates foundational models for visual reasoning, employing computational techniques similar to our objectives, though without relying on programmatic inputs. These collective efforts pave the way for more sophisticated and versatile visual in-context learning frameworks.

## 3 METHOD

### 3.1 UNIFIED REPRESENTATION OF WEATHER UNDERSTANDING TASKS

Weather understanding tasks involve processing multi-source observational data (Veillette et al., 2020), such as geostationary satellites (GEOS), polar-orbiting satellites (POES), weather radars, and ground observation stations. Each task (e.g., weather forecasting, spatial and temporal super-resolution, weather image translation, and post-processing) utilizes different types of input and output data. To address this challenge, we first developed a unified data representation that can standardize these

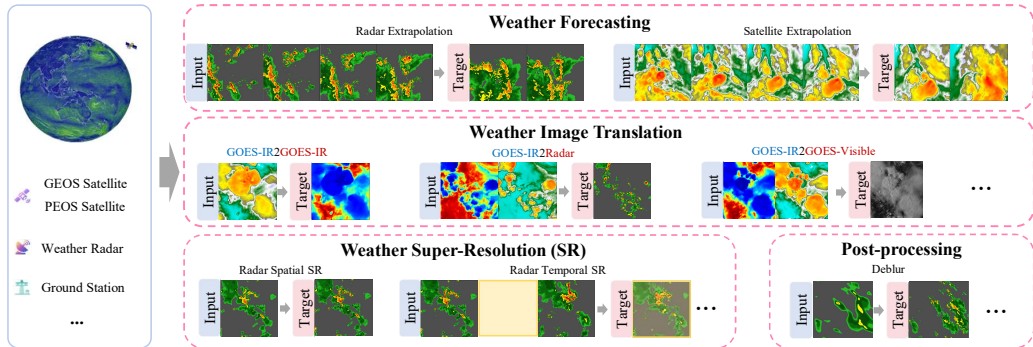

Figure 1: Illustration of the unified representation framework for weather understanding tasks.

diverse tasks. Unlike traditional methods that rely on task-specific models for each distinct task, we introduce a universal foundational model capable of addressing various weather understanding tasks through a single and general solution.

As shown in Figure 1, several key weather understanding tasks can be framed using different types of input and output data. For instance, the weather spatial super-resolution (SR) task generates a high-resolution image $x_{HR}$ from a low-resolution image $x_{LR}$, while weather temporal super-resolution predicts a high-resolution image $x_{HR}^t$ based on two consecutive observed input images $x_{LR}^{t-1}$ and $x_{LR}^{t+1}$, where $t$ represents a particular moment in time. The weather temporal super-resolution task aims to restore the missing observed data in time $t$. Weather forecasting relies on a sequence of observed data points $\{x^1, x^2, \ldots, x^t\}$ that are gathered over the past $t$ time steps. These observed data points serve as condition, enabling the prediction of future data points such as points $\{x^{t+1}, x^{t+2}, \ldots\}$. The image translation task focuses on converting an input image from one modality (e.g., satellite image) to another modality (e.g., radar image). Formally, we can represent these tasks as projections from the source input data $X_S$ to the target output data $X_T$:

$$\tau : X_S \to X_T. \tag{1}$$

When $X_S = x_{LR}$ and $X_T = x_{HR}$, the task corresponds to spatial SR. Similarly, when $X_S = \{x^1, x^2, \ldots, x^t\}$ and $X_T = \{x^{t+1}, x^{t+2}, \ldots\}$, the task represents weather forecasting. As these tasks differ in their input and output formats, as well as sequence lengths, the key challenge lies in unifying them within one coherent data representation.

## 3.2 WEATHERGFM: A WEATHER GENERALIST FOUNDATION MODEL

We present the Weather Generalist Foundation Model (WeatherGFM) to tackle the challenges inherent in a range of weather understanding tasks. Through in-context learning, our WeatherGFM can uniformly handle various weather understanding tasks involving multiple data modalities.

**Weather prompt designing.** In large language models and vision foundation models, task prompts commonly provide specific task-related input-output pairs. As shown in Figure 2, in machine translation (Stahlberg, 2020), the model is given English to French text pairs as prompts. The model can perform machine translation tasks based on these sample prompts for a given input. In visual tasks (Wang et al., 2023a), the visual prompt image1 may be a natural image, and image2 is the corresponding segmented image. The model will conduct the segmentation task for a new input image3 to obtain the segmented image.

Following this paradigm, we designed weather prompts for weather understanding tasks. Since the input for weather understanding tasks involves multiple modalities, such as a single weather observation variable, multiple different weather variables, and time-series weather variables, we proposed three prompts to handle different modalities of input. In Figure 2, weather prompt1 is similar to visual prompts, converting a single modality image into a target image. In weather prompt2, the input modality can be two different channel satellite observation images (e.g., IR069 and IR107 data), and the output can be weather radar observation data for image translation tasks. In weather

**Text Prompt:**   {example: sea otter, loutre de merr}   query: cheese   output: fromage
**Visual Prompt:** {example: image1, image2}   query: image3   output: image4

**Weather Prompts**
**Weather Prompt1:** {example: image1, image2}   query: image3   output: image4
**Weather Prompt2:** {example: image1,image2, image3}   query: image4,image5 output: image6
**Weather Prompt3:** {example: sequence1, sequence2}   query: sequence3   output: sequence4

Figure 2: Comparison of weather prompts with text and visual prompts design.

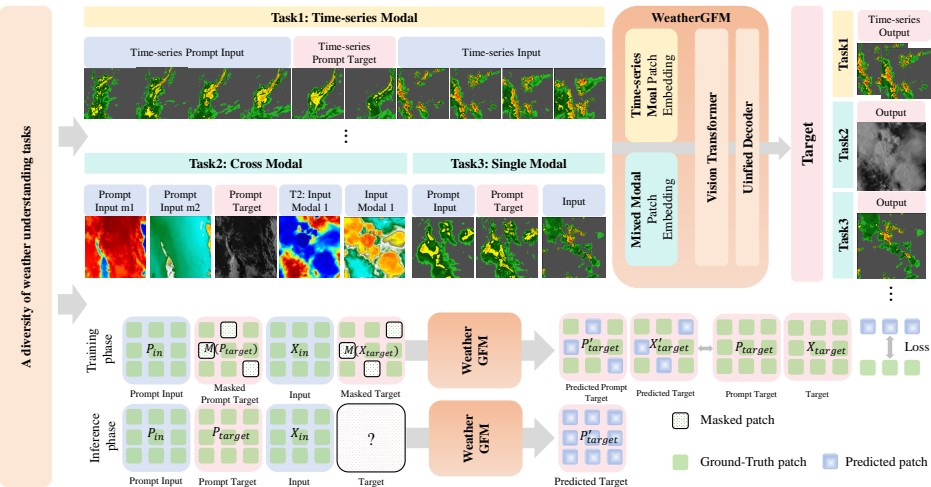

Figure 3: Overall approach of our weather generalist foundation model (WeatherGFM).

prompt3, time-series prompts can be input to perform weather forecasting-related tasks. With these forms of prompt design, our method can handle most weather understanding tasks.

**Weather in-context learning.** Inspired by the success of in-context learning in large language models (Dong et al., 2022) and vision foundation models (Wang et al., 2023a), we propose to unify the weather understanding problem as the visual prompting question-answer paradigm, as illustrated in Eq. 2. Specifically, given a visual question-answer prompt pair $(P_{in}, P_{target})$ as a task-guided prompt and a query input $X_{in}$, the model is expected to perceive the context of the prompt (i.e., what task it represents). Consequently, the model can perform the corresponding operations on the query with the prompt. This process can be formulated as follows:

$$X_{target} = F_\tau(P_{in}, P_{target}, X_{in}; \theta);$$   (2)

where $F_\tau$ represents a universal foundation model parameterized by $\theta$. $P_{in}$ and $P_{target}$ denotes the input and target of task prompts. We can determine what task will be performed on the input $X_{in}$ by selecting the task-specific prompt $P_{in}$ and $P_{target}$, and then obtain the target $X_{target}$ for the corresponding task through the model $F_\tau$.

**Mixed-modal mask modeling.** Upon redefining the output spaces of the aforementioned representative vision tasks, it is observed that both the input and output of these tasks are in the form of images as transformers-based architectures could provide flexibility by treating the image-like data as a set of tokens. Therefore, we build the WeatherGFM architecture on Vision Transformers (ViT) and propose a mixed-modal masked image modeling (MMIM) pipeline to train multiple weather understanding tasks as shown in Figure 3. Inspired by the concept of Visual Question Answering Wang et al. (2023a); Liu et al. (2023b); Chen et al. (2024a), we introduce mixed-modality masking on various weather modalities for visual question-and-answer modeling in weather understanding tasks. This process can be formulated as follows:

$$P'_{target}, X'_{target} = F_\tau(P_{in}, M(P_{target}), X_{in}, M(X_{target}); \theta);$$   (3)

where we randomly conduct mask operation $M$ on the prompt target $P_{target}$ as well as the ground truth $X_{target}$ according to the mask ratio. Meanwhile, the prompt input $P_{in}$ and the input query $X_{in}$

will be retained entirely. $P'_{target}$ and $X'_{target}$ represent the predicted target output of model $F_\tau$. The optimization objectives are as follows:

$$L_\theta^{total} = L_2(P'_{target}, P_{target}) + L_2(X'_{target}, X_{target}). \qquad (4)$$

where we use MSE (mean square error) loss $L_2$ to train the weather generalist foundation model. In the inference stage, we keep the $P_{in}$, $P_{target}$, and $X_{in}$ intact while the target image is fully masked. This target full masking strategy allows generalist foundation models to generate the corresponding target through a visual question-and-answer format. Our WeatherGFM comprises two main elements: the format for input data and the architectural design.

**Input format:** Given an input of shape $(C, H, W)$, ViT predicts an output of shape $(C', H', W')$, where $C$ represents the input channels and $C'$ represents the output channels. As shown in Figure 3, different tasks have different channels. The model tokenizes the input into a sequence of patches, with each patch having a size of $C \times p^2$, where $p$ is the patch size. Unlike RGB-based image data, where the channels are fixed, the number of physical variables in climate and weather data can vary between different datasets and tasks. To adapt the ViT to different weather-related downstream tasks, we designed task-specific patch embedding layers within the architecture. After the patch embedding layer, we use an MLP layer to align the embeddings of different tasks to the same space:

$$\begin{aligned} z_C &= \text{PatchEmbed}_C(x), x \in \mathbb{R}^{C \times H \times W}, \quad z_C \in \mathbb{R}^{N \times D}, \\ z_0 &= \text{MLP}_C(\text{LN}(z_C)), \qquad\qquad\qquad z_0 \in \mathbb{R}^{N \times D} \end{aligned} \qquad (5)$$

where $N, D$ denotes the number of input tokens and the transformer dimension, respectively. For the masked area, we follow previous works (Liu et al., 2023a) to use a learnable token vector to replace each masked patch. We adopt the block-wise masking strategy, taking the masking ratio as 75%.

**Architecture:** A vanilla vision Transformer (ViT) is adopted as the backbone architecture. It consists of task-specific patch-embedded layers and several alternating layers made of Multi-Head Self-Attention (MHSA) and MLP blocks. Layer Normalization (LN) is applied before every block, and residual connections are applied after every block. As shown in Figure 3, this process can be formulated as follows:

$$\begin{aligned} z'_\ell &= \text{MHSA}(\text{LN}(z_{\ell-1})) + z_{\ell-1}, \ell = 1...L, \\ z_\ell &= \text{MLP}(\text{LN}(z'_\ell)) + z'_\ell, \ell = 1...L, \end{aligned} \qquad (6)$$

where L denotes the number of layers. After the attention layers, we employ a prediction head and then unpatchify the output of the prediction head. The prediction head is a one-layer MLP with a hidden dimension of 1024.

## 4 EXPERIMENTS

### 4.1 WEATHER UNDERSTANDING TASKS

We incorporate up to 12 tasks including diverse weather forecasting, weather super-resolution, weather image translation and weather post-processing tasks into our experiments. Specifically, we leverage the Storm EVent ImageRy (SEVIR) (Veillette et al., 2020), ERA5 (Hersbach et al., 2020), POMINO-TROPOMI product (Liu et al., 2020) and GEOS-CF (Keller et al., 2021) datasets to train and evaluate our WeatherGFM. We provide a detailed introduction in Appendix A and B.

### 4.2 IMPLEMENTATION AND EVALUATION

**Training details.** During training, we resize the weather images of different resolutions to a resolution of 256×256 and input them into the model in accordance with the combination mode of $P_{in}, P_{out}, X_{in}, X_{out}$ in the task-specific prompt format, resulting in a N × 256 × 256 total input resolution. The L1 loss is employed as the loss function. For optimization, the AdamW optimizer with a cosine learning rate scheduler is utilized. The base learning rate is 1e4. The batch size is 20.

**Evaluation metrics.** Besides RMSE and ACC, we also include the Critical Success Index (CSI), which is commonly used in weather understanding tasks (e.g., precipitation nowcasting). Details can be found in Appendix C.4

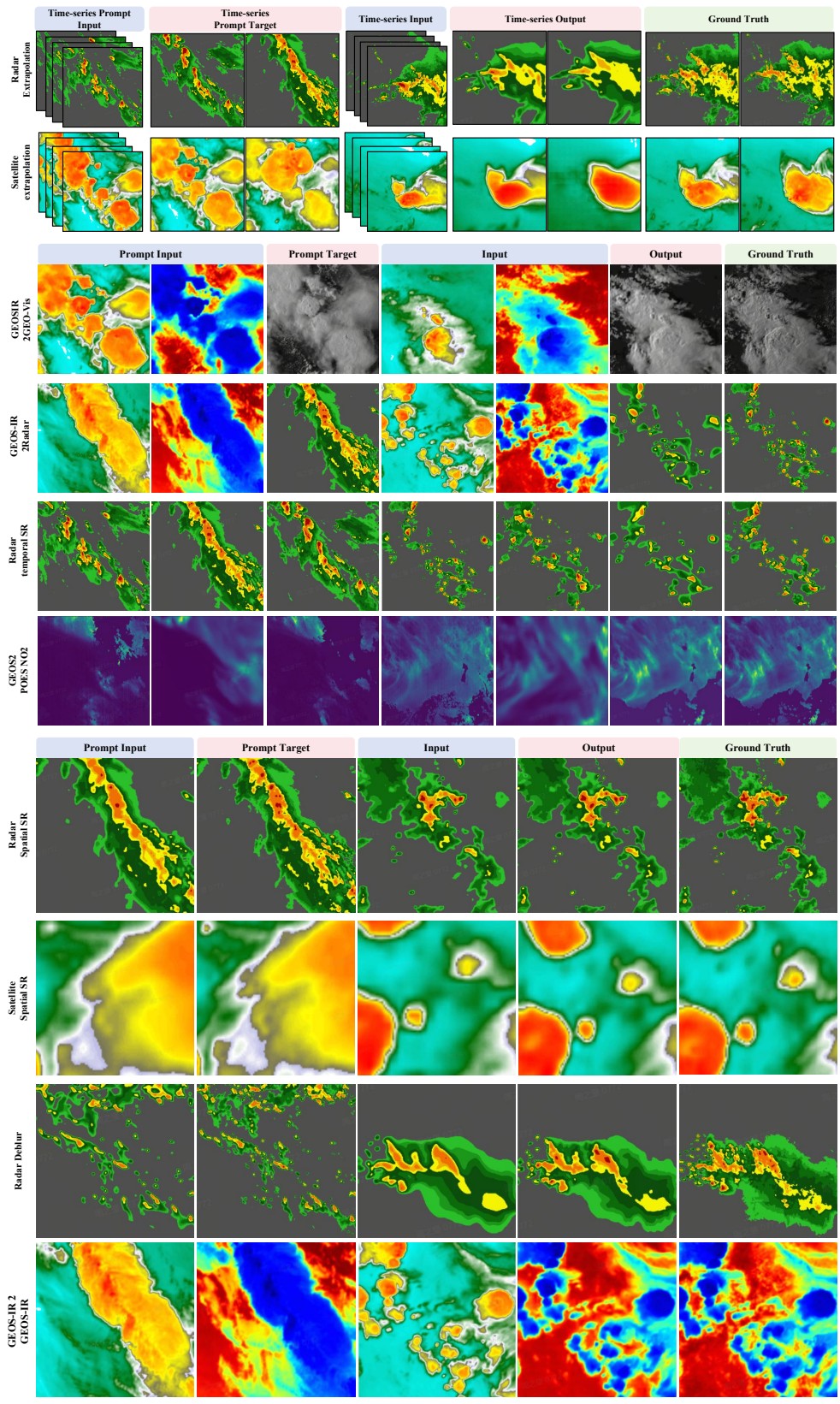

Figure 4: Visual results of the weather understanding tasks by our WeatherGFM.

Table 2: Quantitative results on weather understanding tasks. #: single-task model. †: trained with all weather understanding tasks. ⋆: continual training with REA5 dataset. RMSE, ACC and CSI are calculated as the quantitative metric. A lower RMSE and higher CSI/ACC indicate better results. For Weather Forecasting tasks, we report RMSE scores along with the average ACC score for forecast horizons ranging from 6 to 168 hours. The best results are highlighted in bold, and the second-best results are underscored.

| | Weather super-resolution (SR) | | | | | | | | | |
|---|---|---|---|---|---|---|---|---|---|---|
| Task name | Satellite Spatial SR | | | Radar Temporal SR | | | | Radar Spatial SR | | |
| Metrics | RMSE | CSI/-4000 | CSI/-6000 | RMSE | CSI/74 | CSI/160 | CSI/219 | RMSE | CSI/74 | CSI/160 | CSI/219 |
| UNet# | 0.932 | 0.650 | 0.912 | 0.739 | 0.485 | 0.182 | 0.034 | 0.650 | 0.675 | 0.400 | 0.184 |
| ViT# | _0.047_ | _0.987_ | _0.990_ | _0.333_ | _0.591_ | _0.285_ | _0.061_ | **0.120** | _0.830_ | _0.637_ | _0.358_ |
| WeatherGFM† | **0.042** | **0.988** | **0.996** | **0.327** | **0.597** | **0.287** | **0.073** | _0.121_ | **0.831** | **0.644** | **0.375** |

| | Weather Forecasting | | | | | | | Post-processing | | |
|---|---|---|---|---|---|---|---|---|---|---|
| Task name | Satellite extrapolation | | | Radar extrapolation | | | | Deblur | | |
| Metrics | RMSE | CSI/-4000 | CSI/-6000 | RMSE | CSI/74 | CSI/160 | CSI/219 | RMSE | CSI/74 | CSI/160 | CSI/219 |
| UNet# | 1.033 | 0.617 | 0.900 | 0.815 | 0.353 | _0.082_ | _0.007_ | 0.713 | 0.457 | 0.145 | 0.027 |
| ViT# | _0.408_ | _0.840_ | _0.943_ | _0.490_ | _0.440_ | 0.079 | _0.007_ | **0.163** | _0.594_ | **0.291** | **0.104** |
| WeatherGFM† | **0.347** | **0.863** | **0.951** | **0.467** | **0.465** | **0.128** | **0.021** | _0.264_ | **0.629** | _0.255_ | _0.082_ |

| | Weather image translation | | | | | | | | | |
|---|---|---|---|---|---|---|---|---|---|---|
| Task name | GOES2Radar | | | | | | GOES-IR2GOES-IR | | | |
| Metrics | RMSE | CSI/16 | CSI/74 | CSI/160 | CSI/181 | CSI/219 | RMSE | CSI/-6000 | CSI/-4000 | CSI/0 | CSI/2000 |
| UNet# | 0.821 | 0.222 | 0.370 | _0.180_ | _0.153_ | **0.079** | 0.915 | 0.929 | 0.741 | 0.638 | 0.078 |
| ViT# | _0.445_ | _0.602_ | _0.436_ | _0.180_ | 0.131 | _0.042_ | **0.257** | _0.987_ | **0.972** | **0.809** | _0.136_ |
| WeatherGFM† | **0.436** | **0.619** | **0.447** | **0.208** | **0.157** | 0.053 | _0.310_ | **0.993** | _0.968_ | _0.808_ | **0.222** |

| | GOES-IR2GOES-Visible | | | | | | GOES2POES-$NO_2$ | | | |
|---|---|---|---|---|---|---|---|---|---|---|
| Task name | | | | | | | | | | |
| Metrics | RMSE | CSI/2000 | CSI/3200 | CSI/4400 | CSI/5600 | CSI/6800 | RMSE | CSI/1 | CSI/5 | CSI/10 | CSI/15 |
| UNet# | 0.915 | 0.422 | 0.285 | 0.179 | 0.100 | 0.040 | 0.866 | _0.799_ | 0.360 | 0.274 | _0.202_ |
| ViT# | _0.448_ | _0.574_ | _0.437_ | **0.303** | **0.184** | **0.071** | _0.549_ | **0.841** | _0.432_ | _0.328_ | **0.253** |
| WeatherGFM† | **0.439** | **0.580** | **0.439** | _0.298_ | _0.166_ | _0.068_ | **0.302** | 0.682 | **0.562** | **0.382** | 0.197 |

| | Weather Forecasting T2M | | | | | Weather Forecasting U10 | | | | | |
|---|---|---|---|---|---|---|---|---|---|---|---|
| Task name | | | | | | | | | | | |
| Lead Time [hr.] | 6h | 24h | 72h | 120h | 168h | 6h | 24h | 72h | 120h | 168h | ACC |
| IFS | **0.97** | **1.02** | **1.30** | _1.71_ | 2.23 | **0.79** | **1.11** | **1.92** | 2.89 | 3.81 | _0.836_ |
| ClimaX | 1.11 | _1.19_ | _1.47_ | 1.83 | _2.17_ | _1.04_ | 1.31 | 2.02 | _2.79_ | _3.35_ | 0.824 |
| WeatherGFM⋆ | _1.08_ | 1.23 | 1.56 | **1.68** | **1.76** | 1.12 | _1.26_ | _1.99_ | **2.61** | **3.11** | **0.848** |

Table 3: Standard deviation of the performance computed based on 20 different prompts. Avg. CSI denotes the mean of the CSI score across thresholds [16, 74, 133, 160, 181, 219].

| | GOES2Radar | Radar extrapolation | GOES-IR2GOES-IR | Radar Spatial SR | Radar Temporal SR | Deblur |
|---|---|---|---|---|---|---|
| Avg. RMSE | 0.0087 | 0.0012 | 0.0481 | 0.0001 | 0.0002 | 0.0016 |
| Avg. CSI | 0.0187 | 0.0201 | 0.0284 | 0.0006 | 0.0010 | 0.0047 |

## 4.3 EXPERIMENTAL RESULTS

Currently, there is no general weather foundation model that can comprehensively handle all the discussed weather understanding tasks simultaneously. Although many machine learning methods have been investigated for single tasks, they generally adopt different backbone networks and design strategies tailored to them. For a fair comparison, we have trained a series of baselines (i.e., single-task model) for each weather understanding task under a consistent training setup, including commonly used UNet (Trebing et al., 2021) and ViT (Nguyen et al., 2023) networks. Notably, the purpose of this paper is not to achieve state-of-the-art performance on every task.

**The weather generalist foundation model can achieve strong universal capabilities.** As seen in Table 2, our WeatherGFM, equipped with a straightforward ViT backbone, shows impressive performance and adaptability in 12 weather understanding tasks. It is not only capable of conducting

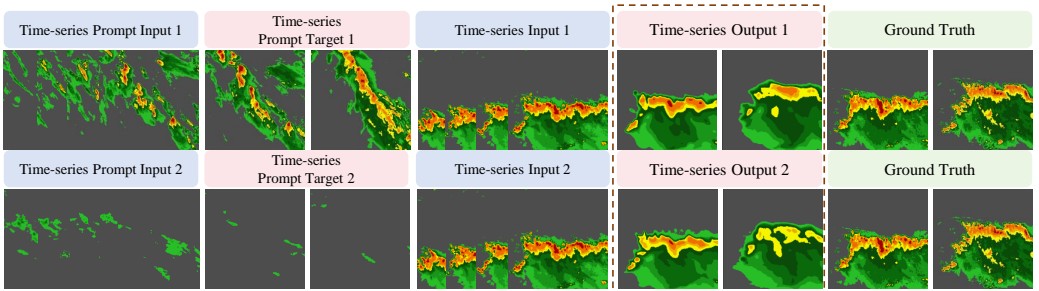

Figure 5: Case studies of our WeatherGFM with different prompts in the radar extrapolation task.

weather forecasting and super-resolution tasks but is also proficient in dealing with weather image translation and post-processing tasks. Overall, our WeatherGFM achieves promising performance on a diversity of weather understanding tasks.

**The weather generalist foundation model outperforms single-task models.** In Table 2, we notice that our WeatherGFM achieves results that outperform the baseline in weather forecasting, weather super-resolution, and image translation tasks. For instance, in radar extrapolation tasks, our WeatherGFM with universal ViT-based model outperforms the single-task ViT model. This indicates that a unified approach to weather understanding tasks can potentially break the performance upperbound of single-task models.

**In-context learning can generate correct outputs across a variety of data modalities and tasks.** As depicted in Figure 4, our WeatherGFM effectively carries out a wide array of weather understanding tasks on multi-modal weather data. In practical scenarios, weather forecasting and weather image transformation represent two substantially different tasks due to differences in temporal modalities. Despite their intricacies, our WeatherGFM with in-context learning can successfully recognize distinct task types, highlighting its significant generalization capacity.

## 4.4 ABLATION STUDIES AND EXPLORATIONS

**Exploration of different task prompts.** To investigate the impact of various visual prompts on quantitative performance, we randomly select 20 meteorological prompts for each task and calculate their quantitative metrics on the test set. Table 3 presents the standard deviation of performance for each task across the 20 distinct meteorological prompts. We note that weather super-resolution tasks are minimally affected by the randomness of weather prompts, whereas weather forecasting tasks and image transformation tasks exhibit more significant variability, reaching approximately 0.02 in CSI. Figure 5 illustrates that for certain weather events, employing different prompts yields more precise outputs. This indicates that our method can comprehend specific weather cases based on weather prompts rather than being a black box model incapable of interactive operations.

**Exploration of generalizability.** To assess the generalizability of our framework, we also utilize the ERA5 dataset (1.40625°) (Hersbach et al., 2020). The WeatherGFM model used for the ERA5 comparison is further trained on the original model by incorporating the ERA5 data, mixed with the previous tasks, as an additional task for continual training. With the introduction of new meteorological variables, we add a new patch embedding layer for each of these variables to the original model. Following the approach of ClimaX (Nguyen et al., 2023), we use a cross-attention module to aggregate all variable embeddings into a single vector. This allows us to handle the task using the single-modal mode in WeatherGFM. Specifically, we employ an MLP layer to align the embeddings for this task. In line with ClimaX (Nguyen et al., 2023), we use 48 ECMWF (European Centre for Medium-Range Weather Forecasts) variables as input and evaluate the performance of WeatherGFM using the temperature at 2 meters above ground (T2m). We consider seven lead times: 6 hours and 1, 3, 5, 7 days, covering a range from nowcasting to short- and medium-range forecasting. Instead of training separate models for each target variable, our WeatherGFM is trained once to predict all variables across all lead times simultaneously. During fine-tuning, we randomize the lead time from 6 hours to 7 days. Table 8 and Table 9 show that our approach significantly outperforms ClimaX in

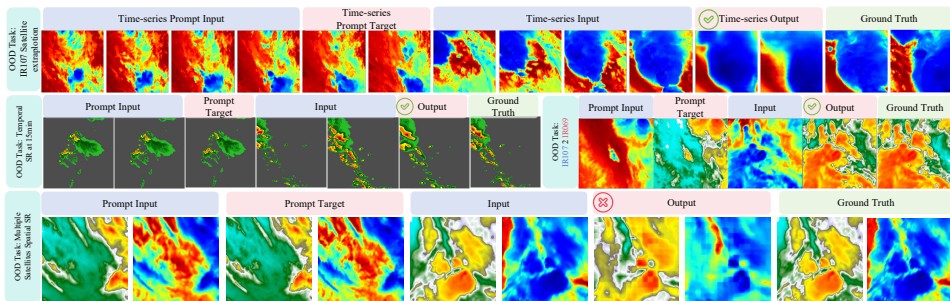

Figure 6: Visual results of our WeatherGFM on OOD tasks.

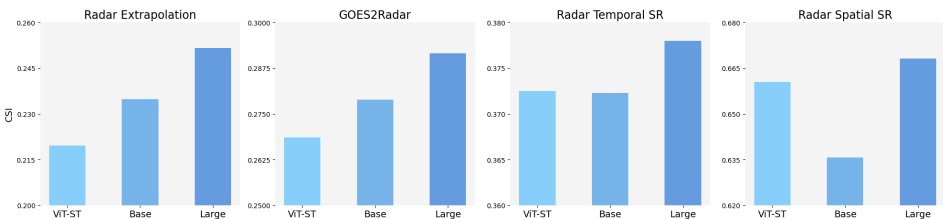

Figure 7: The effect of model sizes. ViT-ST: single-task ViT trained on 0.5 million samples. Base: our WeatherGFM with 100 M parameters trained on 4 million samples. Large: our WeatherGFM with 330 M parameters trained on 4 million samples.

the 120-hour and 168-hour forecast results, even exceeding the ECMWF IFS method. More results can be found in Appendix D.

**Exploration of out-of-distribution tasks.** To evaluate the generalization ability of our WeatherGFM, we have devised a variety of out-of-distribution (OOD) tasks that were not encountered during the training phase, including GEOS-IR107 extrapolation, weather image translation GEOS-IR107 to GEOS-IR069, weather temporal SR at 15 minutes and GEOS-visible satellite extrapolation. As shown in Figure 6, our WeatherGFM generates correct outputs for the first three tasks, which are similar to the training distribution. However, the model encounters difficulties with the more challenging task of multiple-modal satellite spatial SR, where its outputs fail to provide effective meteorological information. These OOD tests demonstrate the model's ability to identify tasks outside the training distribution from new prompts, showcasing a degree of generalization.

**The scaling law for weather foundation models.** To evaluate the impact of data and model scale on performance, we compared single-task models, the base version of our WeatherGFM, and its large version. We established a baseline using a 30M parameter ViT under a single-task with 0.5 million samples. Subsequently, in a multi-task setting with 4 million samples, our model was configured with a base version of 110M and a large version of 330M parameters. Figure 7 illustrates that improvements in performance on various tasks are achieved with the increase of model and data scale. In specific tasks like radar super-resolution, we observe that scaling up both the data and the model is essential for performance gains.

## 5 CONCLUSION

We introduce WeatherGFM, the first generalist foundation model for weather. By utilizing a unified representation across multiple weather understanding tasks and employing a multi-modal prompt design, WeatherGFM effectively addresses a range of tasks, including weather forecasting, super-resolution, image translation, and post-processing through in-context learning. We conduct comprehensive explorations of the model's adaptability to various tasks, its generalization capabilities to unseen tasks, and its scaling behavior with respect to data and model size. We hope this study paves the way for the development of future large-scale generalist foundation models in weather and climate.

ACKNOWLEDGMENTS

This work is supported by Shanghai Artificial Intelligence Laboratory and the China Postdoctoral Science Foundation (Grant No. 2024M761547).

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

## A  DATASETS

**SEVIR.** The Storm EVent ImageRy dataset (SEVIR) (Veillette et al., 2020) is a spatiotemporally aligned dataset that contains over 10,000 weather events represented by five spatially and temporally aligned sensors. These sensors consist of three channels (C02, C09, C13) from the GOES-16 satellite, one NEXRAD derived vertically integrated liquid (VIL) mosaic variable, and lighting detections from the GOES GLM sensor. Each SEVIR event spans 4 hours with 5-minute intervals, sampled randomly (with oversampling of events with moderate and high precipitation) using the NOAA Storm Event Database. In our task, we uniformly resize the resolution of images from different modalities to 256×256. Moreover, we filter the events within the SEVIR dataset and pick out those events that include both the three channels of the GOES-16 satellite and the one variable derived from weather radar. Ultimately, the dataset we utilize comprises 11,508 events with four distinct sensing modalities. Among them, 11,308 events are selected as the training set, while 100 events are designated as the validation set and 100 events are designated as the test set. Consequently, the training set contains a total of 2.2M images, while the validate/test set has a total of 19.6K images. We provide a detailed introduction in Appendix B.

**POMINO-TROPOMI, GEOS-CF.** In addition, we add a weather image translation task for environment monitoring: Translate geostationary $NO_2$ data to polar-orbiting satellites $NO_2$ data (GEOS2POES-$NO_2$) based on POMINO-TROPOMI product (Liu et al., 2020) and GEOS-CF dataset (Keller et al., 2021). In this task, the input images are sourced from GEMS as well as the GEOS-CF datasets, while the output images are obtained from the TROPOMI dataset. The original image has a resolution of 1400×800. We also divide it into grids of 256×256 with a sliding step size of 128. Each original image can thus be segmented into 45 pieces of 256×256 pictures. We utilize the observational data from January 2021 to April 2022. After processing, each modality has 20,000 images with a resolution of 256×256. Among them, we allocate 18,000 images as the training set, 1,000 images as the validation set, and 1,000 images as the test set.

**ERA5.** ERA5 (Hersbach et al., 2020), developed by the European Centre for Medium-Range Weather Forecasts (ECMWF), is a global atmospheric reanalysis dataset that provides detailed information on the Earth's climate and weather conditions from 1940 to the present. It includes a diverse set of variables such as temperature, humidity, precipitation, wind speed and direction, mean sea level pressure, and more. In line with the ClimaX framework, our input selection consists of 48 variables in total: 6 atmospheric variables across 7 vertical levels, 3 surface variables, and 3 constant fields.

## B  DETAILS OF WEATHER UNDERSTANDING TASKS

Table 4: Overview of model inputs, outputs, and prompt formats

| Task | Dataset | Prompt Format | Prompt Input | Prompt Output | Input | Output |
|------|---------|---------------|--------------|---------------|-------|--------|
| Radar Spatial SR | Sevir | Single Modal | VIL LR | VIL HR | VIL LR | VIL HR |
| Satellite Spatial SR | Sevir | Single Modal | IR-069 LR | IR-069 HR | IR-069 LR | IR-069 HR |
| Radar Temporal SR | Sevir | Single Modal | VIL (0,60min) | VIL 30min | VIL (0,60min) | VIL 30min |
| Deblur | Sevir | Single Modal | VIL (Earthformer) | VIL (Ground Truth) | VIL (Earthformer) | VIL (Ground Truth) |
| GEOS-IR2Radar | Sevir | Cross Modal | IR-069,IR-107 | VIL | IR-069,IR-107 | VIL |
| GEOS-IR2GEOS-IR | Sevir | Cross Modal | IR-069 | IR-107 | IR-069 | IR-107 |
| GEOS-IR2GEOS-Vis | Sevir | Cross Modal | IR-069 | VIS | IR-069 | VIS |
| GEOS2POES-NO2 | POMINO | Cross Modal | GEMS, GEOS-CF | TROPOMI | GEMS, GEOS-CF | TROPOMI |
| Satellite extrapolation | Sevir | Time-series Modal | IR-069 (0,30,60,90min) | IR-069 (120,180min) | IR-069 (0,30,60,90min) | IR-069 (120,180min) |
| Radar extrapolation | Sevir | Time-series Modal | VIL (0,30,60,90min) | VIL (120,180min) | VIL (0,30,60,90min) | VIL (120,180min) |
| Weather Forecasting T2M | ERA5 | Cross Modal | 48 variables | t2m variable | 48 variables | t2m variable |
| Weather Forecasting U10 | ERA5 | Cross Modal | 48 variables | U10 variable | 48 variables | U10 variable |

**Weather forecasting.** Radar echo extrapolation aims to forecast data for the subsequent 1-2 hours utilizing observations from past moments (Gao et al., 2024). This task, similar to precipitation nowcasting, plays a significant role in predicting local weather conditions. It can directly impact traffic plans, disaster warnings, and energy management. Likewise, meteorological satellite image extrapolation is crucial for monitoring and analyzing meteorological conditions. Based on the SEVIR dataset, we consider two weather forecasting tasks: radar echo extrapolation Gong et al. (2024) and satellite image extrapolation (Shukla et al., 2011). Our weather prediction tasks incorporate observations from the hour before (0, 30, 60, and 90 minutes past) and the hour ahead (120 and 180 minutes into the future) for both radar and satellite IR-069 extrapolation. Consequently, for this task,

the SEVIR data was extracted and processed to generate 135,696 sequences for training, along with an independent set of 1,200 sequences to validate/test the fitted model.

**Weather super-resolution (SR).** Weather spatial super-resolution task (Veillette et al., 2020) generates a high-resolution image from a low-resolution(LR) image, while temporal super-resolution predicts a high-resolution(HR) image based on two consecutive observed input images. We take into consideration three weather super-resolution tasks: spatial SR for satellite IR-069, spatial SR for radar VIL, and temporal SR for radar VIL with a one-hour interval. We utilize the SEVIR dataset as the source of the HR image. To obtain the LR image, we employ the "Bicubic" interpolation approach, which is commonly used in vision image SR. In the context of meteorology, this is analogous to statistical downscaling, as described in statistical downscaling (Vandal et al., 2017). Specifically, for the VIL image, given that its original resolution is 384×384, we resize it to 256×256 to serve as the HR image and resize it to 64×64 to function as the LR image, thereby implementing a 4x super-resolution task. For the IR-069 image, since its original image has a resolution of 196×196, we resize it to 256×256 to be the HR image and resize it to 64×64 to be the LR image, thus carrying out a 3x super-resolution task. For each spatial SR for satellite tasks, the SEVIR data was extracted and processed to yield 542,784 images for training, along with an independent set of 4,800 images for validating/testing. For the weather temporal SR, we use the radar VIL image at 1 hour (0 and 60 minutes) as the input to predict the radar VIL image at 30 minutes. For the temporal SR task, the SEVIR data was further extracted and processed to generate 407,088 sequences for training, along with an independent set of 3,600 sequences to validate/test the fitted model.

**Weather image translation.** Weather image translation involves converting observation data (e.g., satellite data) to a desired weather image (Veillette et al., 2020). For example, depictions of storms obtained from weather radar are extremely important. However, most areas of the world do not have access to ground-based radar. It is useful for generating weather radar images of storm depictions from satellite observation (Veillette et al., 2020). We consider three weather image translation tasks based on SEVIR dataset: translate geostationary IR-069 to geostationary IR-107 data (GEOS-IR2GEOS-IR), geostationary IR-069 to geostationary Visible data (GEOS-IR2GEOS-Vis), translate geostationary IR-069 and IR-107 to radar VIL data (GEOS-IR2Radar). In addition, we add a weather image translation task for environment monitoring: Translate geostationary $NO_2$ data to polar-orbiting satellites $NO_2$ data (GEOS2POES-$NO_2$) based on POMINO-TROPOMI product (Liu et al., 2020). For the image translation tasks based on SEVIR dataset, we split SEVIR into 542,784 training samples, 4,800 validation samples and 4,800 test samples. For translating geostationary $NO_2$ data,

**Weather post-processing:** Post-processing (e.g., bias correction) aims to minimize or eliminate systematic biases in model outputs and observational data, which emerge due to uncertainties in weather models and measurement errors. Various methods, including statistical, machine learning, and deep learning techniques, can be employed for post-processing, tailoring the approach based on the specific application and data characteristics. By minimizing or eliminating systematic biases, post-processing improves the quality and reliability of weather and climate data. In our experiment, we consider a classic post-processing task: Debluring for radar VIL nowcasting. We employ the output of Earthformer and the corresponding high-quality image as a training sample. Deblurring aims to learn how to map from the output of Earthformer to the corresponding high-quality image.

# C  IMPLEMENTATION DETAILS

## C.1  IMPLEMENTATION DETAILS AND HYPERPARAMETERS

The hyperparameters for WeatherGFM in our experiments is shown in Table refhyperparameters. The L1 loss is employed as the loss function. For optimization, the AdamW optimizer with a cosine learning rate scheduler is utilized. The base learning rate is 1e-4. The batch size is 20 and the accumulation gradient iterations are 4. We use 16 Nvidia A100 GPUs for training. A total of 50 epochs are executed. We leverage fp16 floating point precision in our model.

## C.2  VIT HYPERPARAMETERS

We borrow our ViT implementation from (Beyer et al., 2022). We use the following hyperparameters for ViT in all of our experiments.

Table 5: Default hyperparameters of WeatherGFM

| Hyperparameter | Meaning | Large | Base |
|---|---|---|---|
| $p$ | Patch size | 16 | 16 |
| Encoder dimension | Encoder Embedding dimension | 1024 | 768 |
| Decoder dimension | Decoder Embedding dimension | 512 | 512 |
| Encoder depth | Number of Encoder blocks | 24 | 12 |
| Decoder depth | Number of Encoder blocks | 8 | 8 |
| Encoder Heads | Encoder's attention heads | 16 | 12 |
| Decoder Heads | Decoder's attention heads | 16 | 16 |
| MLP ratio | The hidden dimension of the MLP layer in a ViT block | 4 | 4 |
| Masked ratio | Percentage of the masked target data | 75% | 75% |

Table 6: Hyperparameters of ViT

| Hyperparameter | Meaning | Value |
|---|---|---|
| $p$ | Patch size | 16 |
| Dimension | Embedding dimension | 512 |
| Depth | Number of Encoder blocks | 16 |
| Heads | Encoder's attention heads | 8 |
| MLP dim | Encoder's attention heads | 1024 |

## C.3 UNET HYPERPARAMETERS

We use the following hyperparameters for UNet in all of our experiments.

## C.4 CSI METRIC

The CSI (Critical Success Index) is a commonly used metric used in weather understanding tasks (e.g., precipitation nowcasting). The definition of CSI is:

$$\text{CSI} = \frac{\text{Hits}}{\text{Hits} + \text{Misses} + \text{F.Alarms}}$$

To count the Hits (truth=1, pred=1), Misses (truth=1, pred=0) and F.Alarms (truth=0, pred=1), the prediction and the ground-truth are normalized using mean-variance normalization and binarized at different thresholds. Following SEVIR (Veillette et al., 2020), for radar output tasks, we have established thresholds at [16, 74, 133, 160, 181, 219]. GEOS-visible output tasks are assigned thresholds of [2000, 3200, 4400, 5600, 6800]. The GEOS-IR107 output tasks operate with thresholds set to [-6000, -4000, 0, 2000]. Lastly, the GEOS-IR069 output task employs thresholds of [-4000, -5000, -6000, -7000].

Table 7: Hyperparameters of UNet

| Hyperparameter | Meaning | Value |
|---|---|---|
| Padding size | Padding size of each convolution layer | 1 |
| Kernel size | Kernel size of each convolution layer | 3 |
| Stride | Stride of each convolution layer | 1 |
| Channel multiplications | Number of output channels for Down and Up blocks | [1, 2, 4, 8, 8] |
| Blocks | Number of blocks | 3 |
| Use attention | If use attention in Down and Up blocks | False |
| Dropout | Dropout rate | 0 |
| Inner channel | Number of channels in the intermediate layers | 64 |

# D EXTENDABILITY

Tables 8 and Table 9 show the comparison of RMSE and ACC with different lead times for IFS, ClimaX, and our WeatherGFM on the variables t2m and U10. Our method achieves comparable results to ClimaX at other lead times. It's worth noting that Climax fine-tunes the pre-train model for each lead time, meaning Climax requires N models for weather forecasting at N lead times. In contrast, our universal model can handle all lead time tasks with just a single model, without the need for task-specific fine-tuning. Moreover, the Climax method trained for 100 epochs using 80 V100 GPUs, while our generalist model trained for 20 epochs using 8 A100 GPUs. This indicates that our generalist model converges much faster than Climax. On the other hand, Aurora, focusing on forecasting tasks, employs resolutions of 0.25° and 0.1°, which often produce better quantitative results than 1.4°. For comparison with ClimaX, we selected the 1.4° setting for our atmospheric forecasting experiments. Our generalist model has outperformed ClimaX on multiple variables in these experiments.

Table 8: Comparison of RMSE and ACC across different lead times for IFS, ClimaX, and ours WeatherGFM on t2m variable. ClimaX views predicting at each lead time as a separate task and fine-tunes a separate model for every individual task. In contrast, our WeatherGFM utilizes a single model to deal with all of these tasks. Aurora is a forecasting foundation model that is trained using higher resolution atmospheric data from ERA5 (0.25°), making its results not directly comparable.

| Lead Time | RMSE ↓ | | | | ACC ↑ | | |
|---|---|---|---|---|---|---|---|
| [hr.] | IFS | Aurora (0.25°) | ClimaX | WeatherGFM | IFS | ClimaX | WeatherGFM |
| 6 | **0.97** | 0.53 | 1.11 | 1.08 | **0.99** | 0.98 | 0.98 |
| 24 | **1.02** | 0.68 | 1.19 | 1.23 | **0.99** | 0.97 | 0.97 |
| 72 | **1.30** | 0.96 | 1.47 | 1.56 | **0.98** | 0.96 | 0.96 |
| 120 | 1.71 | 1.32 | 1.83 | **1.68** | **0.96** | 0.94 | 0.95 |
| 168 | 2.23 | 1.73 | 2.17 | **1.76** | 0.93 | 0.91 | **0.94** |

Table 9: Comparison of RMSE and ACC across different lead times for IFS, ClimaX,and ours WeatherGFM on U10 variable. WeatherGFM utilizes a single model to deal with different lead time tasks.

| Lead Time | RMSE ↓ | | | | ACC ↑ | | |
|---|---|---|---|---|---|---|---|
| [hr.] | IFS | Aurora (0.25°) | ClimaX | WeatherGFM | IFS | ClimaX | WeatherGFM |
| 6 | **0.79** | 0.69 | 1.04 | 1.12 | **0.98** | 0.97 | 0.97 |
| 24 | **1.11** | 0.97 | 1.31 | 1.26 | **0.97** | 0.95 | 0.95 |
| 72 | **1.92** | 1.56 | 2.02 | 1.99 | **0.89** | 0.87 | 0.88 |
| 120 | 2.89 | 2.27 | 2.79 | **2.61** | 0.76 | 0.74 | **0.79** |
| 168 | 3.81 | 2.98 | 3.35 | **3.11** | 0.58 | 0.59 | **0.65** |

# E EFFECTS OF MULTI-TASK TRAINING

In order to assess the influence of multi-task training on performance, we contrasted the versions of our WeatherGFM that were trained on 4 tasks and 10 tasks respectively. Additionally, for the purpose of comparison with WeatherGFM, we also made use of the single-task Vision Transformer (ViT). As shown in the table 12, WeatherGFM-4tasks is trained on four tasks: Radar Temporal SR, GOES2GOES, GOES2Radar, and Radar Spatial SR. It uses more data and encompasses more tasks than the ViT-ST specialized model, which is trained on these four tasks separately. However, WeatherGFM-4tasks utilizes less data and fewer tasks compared to WeatherGFM-10tasks, which is trained on ten tasks. The results indicate that for radar image generation tasks (Radar Temporal SR, GOES2Radar, and Radar Spatial SR), both WeatherGFM-4tasks and WeatherGFM-10tasks outperform ViT-ST. This is likely because most of the selected tasks are related to radar image generation. However, for the satellite image generation task (GOES2GOES), WeatherGFM-4tasks

Table 10: Influence of employing different visual prompts on different tasks. The color red is used for the poorest-performing prompts, and green is used for the best-performing prompts.

| Prompts | | Idx0 | Idx1 | Idx2 | Idx3 | Idx4 | Idx5 | Idx6 | Idx7 | Idx8 | Idx9 |
|---|---|---|---|---|---|---|---|---|---|---|---|
| GOES2Radar | RMSE | 0.4759 | 0.471 | 0.5117 | 0.4748 | 0.4691 | 0.4725 | 0.4726 | 0.4692 | 0.4722 | 0.473 |
| | CSI | 0.3401 | 0.3335 | 0.3233 | 0.3300 | 0.3452 | 0.3298 | 0.3272 | 0.2467 | 0.3302 | 0.3317 |
| Radar Extrapolation | RMSE | 0.4912 | 0.4908 | 0.4891 | 0.4925 | 0.4933 | 0.4937 | 0.4933 | 0.493 | 0.4933 | 0.4951 |
| | CSI | 0.3401 | 0.2497 | 0.2534 | 0.2479 | 0.2484 | 0.2467 | 0.2462 | 0.2467 | 0.2453 | 0.2467 |
| GOES-IR2GOES-IR | RMSE | 0.283 | 0.2947 | 0.2593 | 0.3791 | 0.2424 | 0.2989 | 0.3213 | 0.2811 | 0.4372 | 0.387 |
| | CSI | 0.6851 | 0.6907 | 0.6963 | 0.6559 | 0.6958 | 0.7525 | 0.6835 | 0.7314 | 0.6240 | 0.6529 |
| Radar Spatial SR | RMSE | 0.1331 | 0.1331 | 0.1334 | 0.1331 | 0.1332 | 0.1335 | 0.1334 | 0.1333 | 0.1333 | 0.1335 |
| | CSI | 0.7208 | 0.7206 | 0.7209 | 0.7208 | 0.7228 | 0.7222 | 0.7221 | 0.7227 | 0.7217 | 0.7225 |
| Radar Temporal SR | RMSE | 0.3166 | 0.3166 | 0.3164 | 0.3164 | 0.3168 | 0.3172 | 0.3168 | 0.3166 | 0.317 | 0.3172 |
| | CSI | 0.3387 | 0.3395 | 0.3389 | 0.3359 | 0.3398 | 0.3366 | 0.3376 | 0.3376 | 0.3372 | 0.3388 |
| Deblur | RMSE | 0.2272 | 0.2236 | 0.2236 | 0.2255 | 0.2253 | 0.2274 | 0.2273 | 0.2286 | 0.2241 | 0.2233 |
| | CSI | 0.3412 | 0.3405 | 0.3420 | 0.3413 | 0.3407 | 0.3406 | 0.3408 | 0.3408 | 0.3410 | 0.3419 |
| Prompts | | Idx10 | Idx11 | Idx12 | Idx13 | Idx14 | Idx15 | Idx16 | Idx17 | Idx18 | Idx19 |
| GOES2Radar | RMSE | 0.4711 | 0.4762 | 0.4731 | 0.4706 | 0.4743 | 0.4752 | 0.4747 | 0.4764 | 0.4735 | 0.4725 |
| | CSI | 0.3260 | 0.3252 | 0.3277 | 0.3266 | 0.3264 | 0.3297 | 0.3248 | 0.3247 | 0.3276 | 0.3265 |
| Radar Extrapolation | RMSE | 0.4927 | 0.4931 | 0.4934 | 0.493 | 0.4938 | 0.4929 | 0.4932 | 0.4934 | 0.4937 | 0.4934 |
| | CSI | 0.2478 | 0.2474 | 0.2483 | 0.2483 | 0.2493 | 0.2488 | 0.2474 | 0.2477 | 0.2479 | 0.2474 |
| GOES-IR2GOES-IR | RMSE | 0.4052 | 0.3017 | 0.2917 | 0.3136 | 0.3027 | 0.2828 | 0.2886 | 0.3189 | 0.3147 | 0.3102 |
| | CSI | 0.6451 | 0.7057 | 0.7034 | 0.6930 | 0.7064 | 0.7044 | 0.7076 | 0.6899 | 0.6982 | 0.7040 |
| Radar Spatial SR | RMSE | 0.1332 | 0.1333 | 0.1332 | 0.1332 | 0.1332 | 0.1332 | 0.1332 | 0.1333 | 0.1332 | 0.1332 |
| | CSI | 0.7222 | 0.7221 | 0.7215 | 0.7216 | 0.7217 | 0.7219 | 0.7218 | 0.7215 | 0.7224 | 0.7213 |
| Radar Temporal SR | RMSE | 0.3166 | 0.3168 | 0.3167 | 0.3168 | 0.3166 | 0.3167 | 0.3166 | 0.3169 | 0.317 | 0.3166 |
| | CSI | 0.3359 | 0.3374 | 0.3370 | 0.3371 | 0.3381 | 0.3367 | 0.3375 | 0.3374 | 0.3371 | 0.3376 |
| Deblur | RMSE | 0.2252 | 0.2233 | 0.2272 | 0.2257 | 0.2259 | 0.2257 | 0.2264 | 0.2267 | 0.2235 | 0.2263 |
| | CSI | 0.3405 | 0.3405 | 0.3413 | 0.3417 | 0.3416 | 0.3418 | 0.3416 | 0.3417 | 0.3414 | 0.3415 |

Table 11: Comparison of the models with random prompts, high-quality prompts, and searched prompts according to RMSE.

| Tasks | GOES2Radar | | | | | Radar Extrapolation | | | | |
|---|---|---|---|---|---|---|---|---|---|---|
| Metrics | CSI/74 | CSI/133 | CSI/160 | CSI/181 | CSI/219 | CSI/74 | CSI/133 | CSI/160 | CSI/181 | CSI/219 |
| random prompts | 0.389 | 0.238 | 0.194 | 0.15 | 0.048 | 0.423 | 0.187 | 0.106 | 0.069 | 0.017 |
| high prompts | 0.399 | 0.249 | 0.208 | 0.164 | 0.057 | 0.426 | 0.191 | 0.11 | 0.072 | 0.019 |
| searched prompts | 0.401 | 0.24 | 0.199 | 0.155 | 0.049 | 0.425 | 0.188 | 0.108 | 0.071 | 0.018 |

Table 12: The effect of multi-task training. ViT-ST: single-task ViT. WeatherGFM-4Tasks: our WeatherGFM trained on 4 tasks. WeatherGFM-10Tasks: our WeatherGFM trained on all tasks.

| Tasks | GOES2Radar | | | | | Radar Temporal SR | | | | |
|---|---|---|---|---|---|---|---|---|---|---|
| Metrics | RMSE | CSI/74 | CSI/133 | CSI/181 | CSI/219 | RMSE | CSI/74 | CSI/133 | CSI/181 | CSI/219 |
| ViT-ST | 0.445 | 0.424 | 0.242 | 0.134 | 0.045 | 0.333 | 0.585 | 0.366 | 0.215 | 0.063 |
| WeatherGFM-4Tasks | 0.460 | 0.443 | 0.263 | 0.166 | 0.059 | 0.353 | 0.576 | 0.355 | 0.209 | 0.074 |
| WeatherGFM-10Tasks | 0.436 | 0.447 | 0.266 | 0.157 | 0.053 | 0.327 | 0.597 | 0.376 | 0.217 | 0.073 |
| Tasks | Radar Spatial SR | | | | | GOES2GOES | | | | |
| Metrics | RMSE | CSI/74 | CSI/133 | CSI/181 | CSI/219 | RMSE | CSI/-6K | CSI/-4K | CSI/0 | CSI/2K |
| ViT-ST | 0.120 | 0.820 | 0.703 | 0.573 | 0.387 | 0.257 | 0.987 | 0.972 | 0.809 | 0.136 |
| WeatherGFM-4Tasks | 0.120 | 0.831 | 0.714 | 0.574 | 0.380 | 0.317 | 0.994 | 0.968 | 0.766 | 0.148 |
| WeatherGFM-10Tasks | 0.121 | 0.831 | 0.712 | 0.570 | 0.375 | 0.310 | 0.993 | 0.968 | 0.808 | 0.222 |

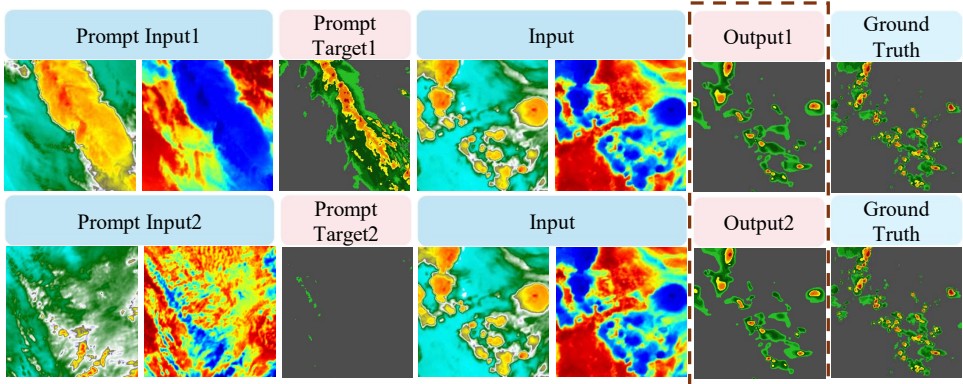

Figure 8: Case studies of our WeatherGFM using different prompts in the GEOS-IR2Radar task.

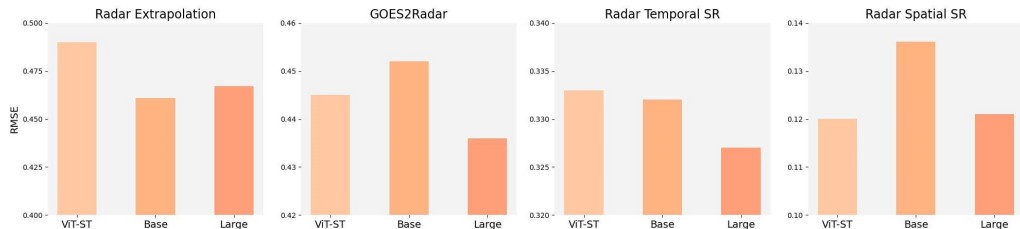

Figure 9: RMSE performance comparison across different model configurations.

does not perform as well. This suggests that multi-task learning of similar tasks can enhance the model's performance on those tasks.

## F MORE DETAILS OF SCALING LAW

In Figure 9, we present the results of the RMSE metric that compares single-task models, the base version of our WeatherGFM, and its large version. It can be observed that increasing the capacity of the model generally leads to better performance when the data size remains constant. In contrast, for smaller models, an increase in training data may result in poorer performance. We hypothesize that this could be due to the specificity of different tasks within the training data, which makes it more challenging for the model to fit effectively.

## G EFFECTS OF VISUAL PROMPTS

To assess the effectiveness of visual prompts within the WeatherGFM framework, we conducted a series of tests across various scenarios, including GOES2Radar, Radar Extrapolation, GOES-IR2GOES-IR, Radar Spatial SR, Radar Temporal SR, and Deblur tasks. In our main paper, we detail a thorough process of prompt selection, curating 20 unique prompts for each task and subsequently reporting the most favorable quantitative outcomes. The comprehensive results of these prompt variations on several representative tasks are presented in the tables. By examining Table 10, we observed that prompts have a significant impact on the GOES2Radar and Radar Extrapolation tasks. Consequently, we conducted further experiments on these two tasks. The results are shown in Table 11. In these experiments, "random prompt" refers to prompts selected randomly, while "high prompt" refers to prompts selected from a high-quality prompt base derived from radar data. Specifically, we grouped 100 events and selected samples that contained values exceeding a threshold of 50 within each group, totaling 113 samples. Prompts were then randomly selected from this high-quality base for the experiments. "Searched prompt" refers to a method where prompts are selected based on the RMSE metric calculated from the input images to find similar prompts.

Table 13: Quantitative evaluation on OOD tasks. * denotes that WeatherGFM has not been trained or fine-tuned for the tasks listed below and conducts generalized inference directly. # indicate that UNet and ViT undergo supervised training on the corresponding training dataset.

| Tasks | IR107 Satellite extraplotion | | | | IR107 2 IR069 | | |
|---|---|---|---|---|---|---|---|
| Metrics | RMSE | CSI/-4K | CSI/0 | CSI/2K | RMSE | CSI/-4K | CSI/-6K |
| UNet# | 0.991 | 0.695 | 0.642 | 0.074 | 0.942 | 0.642 | 0.910 |
| ViT# | 0.413 | 0.899 | 0.776 | 0.245 | 0.212 | 0.958 | 0.986 |
| WeatherGFM* | 0.389 | 0.903 | 0.774 | 0.244 | 0.340 | 0.934 | 0.986 |
| Tasks | Temporal SR at 15min | | | | | | |
| Metrics | RMSE | CSI/16 | CSI/74 | CSI/133 | CSI/160 | CSI/181 | CSI/219 |
| UNet# | 0.676 | 0.211 | 0.627 | 0.428 | 0.351 | 0.262 | 0.083 |
| ViT# | 0.218 | 0.838 | 0.761 | 0.598 | 0.525 | 0.445 | 0.190 |
| WeatherGFM* | 0.272 | 0.814 | 0.703 | 0.507 | 0.419 | 0.336 | 0.117 |

Table 14: More quantitative results on weather understanding tasks. We also report the results of the POD and FAR metrics.

| | Downscaling | | | | | | Forecasting | |
|---|---|---|---|---|---|---|---|---|
| Task name | Satellite Spatial SR | | Radar Temporal SR | | Radar Spatial SR | | Radar Extrapolation | |
| Metrics | AVG. POD | AVG. FAR | AVG. POD | AVG. FAR | AVG. POD | AVG. FAR | AVG. POD | AVG. FAR |
| UNet | 1.0000 | 0.7466 | 0.4380 | 0.1738 | 0.6568 | 0.1702 | 0.3207 | 0.1751 |
| ViT | 0.9941 | 0.0560 | 0.4520 | 0.0139 | 0.7219 | 0.0047 | 0.2749 | 0.0228 |
| WeatherGFM | 0.9973 | 0.05400 | 0.4362 | 0.0111 | 0.7320 | 0.0039 | 0.3028 | 0.01620 |
| | Inversion | | | | | | Forecasting | |
| Task name | GOES2Radar | | GOES-IR2GOES-Visible | | GOES2POES-NO2 | | Satellite Extrapolation | |
| Metrics | AVG. POD | AVG. FAR | AVG. POD | AVG. FAR | AVG. POD | AVG. FAR | AVG. POD | AVG. FAR |
| UNet | 0.4691 | 0.1821 | 0.4260 | 0.1358 | 0.5993 | 0.2548 | 1.0000 | 0.7841 |
| ViT | 0.3537 | 0.0229 | 0.4626 | 0.0829 | 0.5421 | 0.1686 | 0.9599 | 0.3514 |
| WeatherGFM | 0.3524 | 0.0147 | 0.3630 | 0.1388 | 0.5023 | 0.0024 | 0.9578 | 0.2379 |

## H  ADDITIONAL QUANTITATIVE RESULTS

In this section, we present more quantitative results on various weather understanding tasks. Table 13 provides quantitative evaluation on OOD tasks. Table 14 provides a detailed comparison of these metrics for the UNet, ViT, and WeatherGFM models. AVG.POD and AVG.FAR represent the mean scores across different thresholds for various tasks. For radar output tasks, the thresholds are [16, 74, 133, 160, 181, 219]; for GEOS-visible output tasks, the thresholds are [2000, 3200, 4400, 5600, 6800]; and for GEOS-IR069 output tasks, the thresholds are [-4000, -5000, -6000, -7000].

## I  MORE VISUAL RESULTS

To comprehensively assess the performance of WeatherGFM, we present a range of qualitative visual results across various tasks, including weather forecasting, weather super-resolution, and weather image translation. Additionally, we conduct a comparative evaluation against the unified ViT-large model, as well as single-task ViT and UNet models. The visual outputs and comparisons are thoughtfully illustrated in Figure 10. WeatherGFM's proficiency in generating visually appealing outputs is readily evident in the presented results. Notably, the visual quality surpasses that of the baseline models. However, the significance of WeatherGFM's capability extends beyond visual quality. Its distinctive strength lies in its ability to effectively handle a wide array of image enhancement tasks and image detection. This marks a noteworthy distinction from traditional models, which often struggle to concurrently address such a diverse spectrum of tasks.

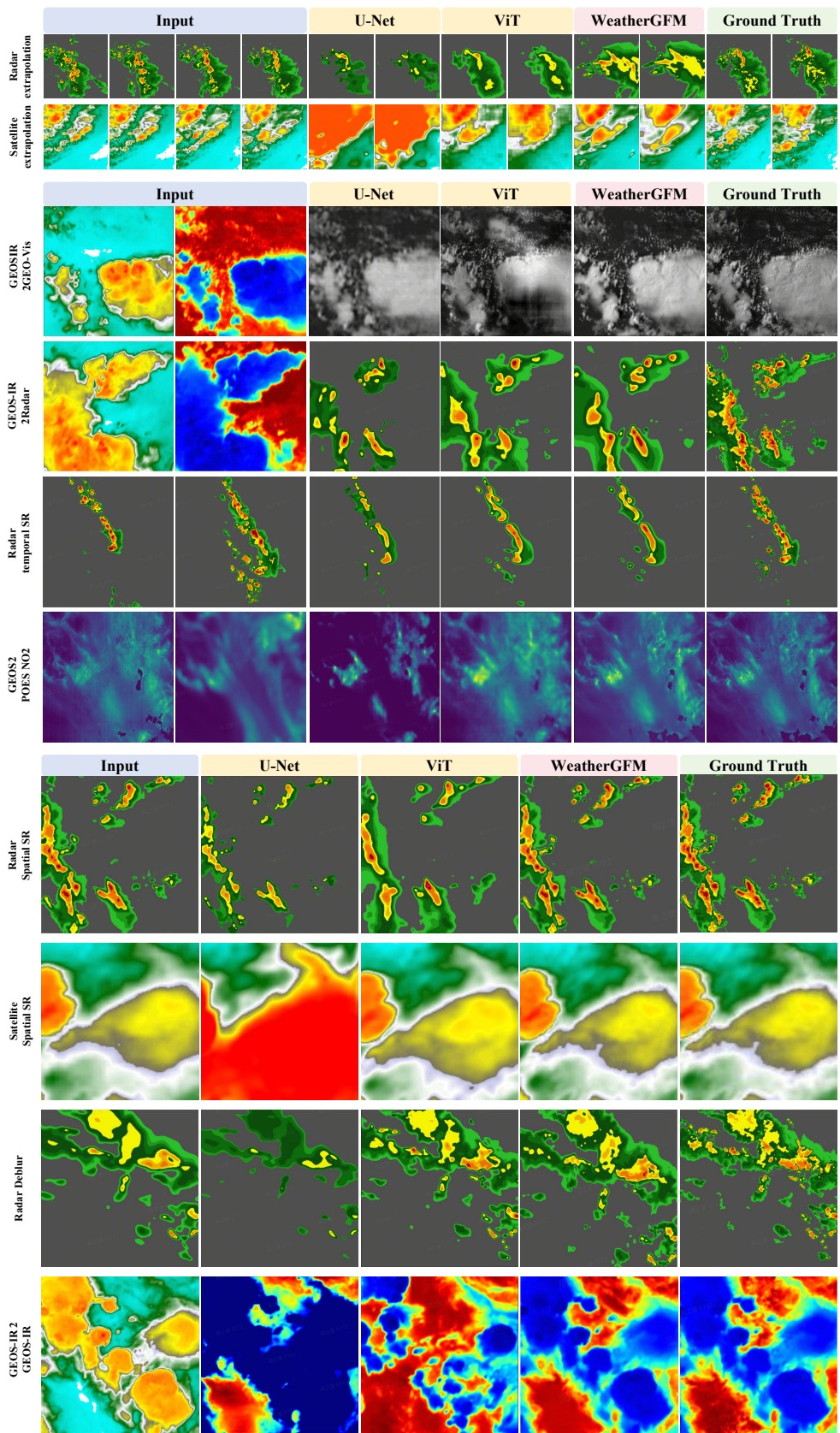

Figure 10: Visual results of the weather understanding tasks by our WeatherGFM.

