# OpenReview forum: "WeatherGFM: Learning a Weather Generalist Foundation Model via In-context Learning"
_ICLR.cc/2025/Conference — ICLR 2025 Poster_

### Official Review · Reviewer_qe7B · 2024-10-27

**Soundness:** 2
**Presentation:** 4
**Contribution:** 4
**Rating:** 10
**Confidence:** 5

**Summary:**

The paper presents WeatherGFM, a pioneering general-purpose weather foundation model that unifies diverse weather-related tasks through contextual learning. By integrating different weather data formats, WeatherGFM is trained with visual prompts to handle a wide range of tasks, including forecasting, super-resolution, image translation, and post-processing. The model is designed to overcome the limitations of task-specific models by leveraging a single architecture capable of understanding and adapting to varying weather data patterns. Extensive experiments show that WeatherGFM achieves state-of-the-art performance across multiple tasks and exhibits impressive generalization on unseen tasks, demonstrating its potential as a robust, adaptable solution for complex weather modeling challenges.

**Strengths:**

1.	The proposed WeatherGFM can handle different weather data modalities and task objectives in a unified manner
2.	In the unseen understanding task, the WeatherGFM still shows a certain generalization.
3.	The introduction of the model input format is very clear, and the classification and identification of various weather image tasks are fully explained

**Weaknesses:**

1.	A mixed-modal masked image modeling (MMIM) pipeline, which is the core innovation part of the proposed framework of WeatherGFM, has not been introduced in this paper. Other parts of  WeatherGFM architecture are actually closer to the comparative ClimaX model framework. It will lack the innovative elaboration of the model framework.
2.	CSI is the main evaluation index in this paper, but ACC is also a very important index to evaluate the correlation of weather. Should this index be taken into account? Moreover, the setting of CSI threshold value needs some explanation in this paper to explain its important reference for this kind of weather situation classification.
3.	According to the comparative results of Table 2 experiments in this paper, in fact, the basic Vision Transformer can also realize a variety of understanding tasks of various data sets in the form of input format proposed by the author, and the effect of some tasks is better than that of WeatherGFM, so the advantages of this model in multi-task are specifically highlighted.
4.	There are two clerical errors: (a) "the" should be capitalized. (b) The basic learning rate in line 366 should be 1e-4.

**Questions:**

1. Can you explain in detail which part of the module WeatherGFM shows its advantages in various tasks?
2. Can you simply explain the significance of CSI threshold setting?

---

> ### Author Response · Authors · 2024-11-24
> **Response to Reviewer qe7B (Part 1)**
>
> Dear Reviewer qe7B,
>
> We appreciate your thoughtful and positive feedback. We have responded to the concerns mentioned in your comments below and hope that our clarifications will further strengthen your confidence in our work.
>
> #### W1. The introduction of mixed-modal masked image modeling (MMIM) pipeline.
>
> Thank you for your suggestions. we have made detailed revisions in the main paper, please refer to page 5. The mixed-modal masked image modeling process can be formulated as follows:
> $$
> P' _ {\text{target}}, X' _ {\text{target}} = F _ {\tau} (P _ {\text{in}}, M(P _ {\text{target}}), X _ {\text{in}}, M(X _ {\text{target}}); \theta);
> $$
> where we randomly conduct mask operation $M$ on the prompt target $P _ {target}$ as well as the ground truth $X _ {target}$ according to the mask ratio. Meanwhile, the prompt input $P _ {in}$ and the input query $X _ {in}$ will be retained entirely.  $P^{'} _ {target}$ and $X^{'} _ {target}$ represent the predicted target output of model $F _ {\tau}$.
> The optimization objectives are as follows:
> $$
> \mathrm{L}^{total} _ {\theta} = \mathrm{L} _ 2(P^{'} _ {target},P _ {target}) + \mathrm{L} _ 2(T^{'} _ {target},T _ {target}).
> $$
> where we use MSE (mean square error) loss $\mathrm{L _ 2}$ to train the weather generalist foundation model. In the inference stage, we keep the $P _ {in}$, $P _ {out}$, and $X _ {in}$ intact while the target image is fully masked. This target full masking strategy allows generalist foundation models to generate the corresponding target through a visual question-and-answer format.
>
> #### W2. Evaluation Metrics
>
> >2.1 Should ACC be taken into account?
>
> * Given that our current training and testing data are primarily based on the SEVIR dataset, which includes a large number of storm-related events, it is more appropriate to use the Threat Score/Critical Success Index (CSI) as a metric for evaluating extreme events. Previous works, such as SEVIR [1], CasCast [2], and EarthFormer [3], typically use CSI as a primary evaluation metric. The Anomaly Correlation Coefficient (ACC) is also an important metric in weather forecasting. It assesses the quality of a forecasting system by calculating the correlation between forecasts and observations. This metric is suitable for evaluating global medium-range forecast datasets, such as ERA5. In our extended experiments, we used the ERA5 dataset for testing and employed RMSE and ACC as evaluation metrics.
>
> >2.2 Can you simply explain the significance of CSI threshold setting?
> >
> * For the CSI threshold settings of radar images, we followed SEVIR [1] and chose thresholds based on the six Video Integrator and Processor (VIP) intensity levels, corresponding to pixel values [16, 74, 133, 160, 181, 219]. Similarly, for satellite images, we selected representative thresholds based on the level information provided in SEVIR dataset for different variables.
>
> [1] Veillette M, Samsi S, Mattioli C. Sevir: A storm event imagery dataset for deep learning applications in radar and satellite meteorology[J]. Advances in Neural Information Processing Systems, 2020, 33: 22009-22019.
> [2] Gong J, Bai L, Ye P, et al. Cascast: Skillful high-resolution precipitation nowcasting via cascaded modelling[J]. arXiv preprint arXiv:2402.04290, 2024.
> [3] Gao Z, Shi X, Wang H, et al. Earthformer: Exploring space-time transformers for earth system forecasting[J]. Advances in Neural Information Processing Systems, 2022, 35: 25390-25403.

---

> ### Author Response · Authors · 2024-11-24
> **Response to Reviewer qe7B (Part 2)**
>
> #### W3. Advantages of this model in multi-task
> > The basic Vision Transformer can also realize a variety of understanding tasks of various data sets in the form of inAput format proposed by the author, and the effect of some tasks is better than that of WeatherGFM.
> * Our WeatherGFM have achieved siginificant performance improvment than specialized models on most tasks, such as the satellite forecasting, radar satellite forecasting, GOES2Radar translation and etc. It is noteworthy that we employ a single model to handle 10 tasks instead of using 10 specialised models for 10 tasks.
> * In fact, different tasks have varying levels of learning difficulty. Some simple tasks are hard to improve further because they are easier to learn to a high performance level. This phenomenon can also be observed in existing works[2,3,4]. In our experiments, the GOES-IR2GOES-IR image translation task is simpler compared to other tasks. For example, the CSI index for the GOES-IR2GOES-IR task has reached around 0.99. Our model has achieved sufficiently good performance.
> * The most critical advantages of our method over specialized models is its versatility and generalization ability. Specialized model data-driven models struggle to generalize to new tasks, while our model shows versatility across multiple tasks and generalization ability on unseen tasks without any fine-tuning. This is an important attribute of AI-based weather foundation models, as mentioned in existing works[1,5].
>
>
> [1] Nguyen T, Brandstetter J, Kapoor A, et al. ClimaX: A foundation model for weather and climate[J]. arXiv preprint arXiv:2301.10343, 2023.
>
> [2] Wang X, Wang W, Cao Y, et al. Images speak in images: A generalist painter for in-context visual learning[C]//Proceedings of the IEEE/CVF Conference on Computer Vision and Pattern Recognition. 2023: 6830-6839.
>
> [3] Liu Y, Chen X, Ma X, et al. Unifying image processing as visual prompting question answering[J]. arXiv preprint arXiv:2310.10513, 2023.
>
> [4] Chen X, Liu Y, Pu Y, et al. Learning a low-level vision generalist via visual task prompt[C]//Proceedings of the 32nd ACM International Conference on Multimedia. 2024: 2671-2680.
>
> [5] Chen S, Long G, Jiang J, et al. Foundation models for weather and climate data understanding: A comprehensive survey[J]. arXiv preprint arXiv:2312.03014, 2023.
>
>
> #### W4. Can you explain in detail which part of the module WeatherGFM shows its advantages in various tasks?
>
> The in-context learning strategy of our WeatherGFM shows advantages in various tasks.
> * Compared with existing foundation models (e.g., ClimaX), the advantages of our model lie in its versatile ability ability to learn various tasks through in-context learning. Currently, foundation models are usually pre-trained on prediction tasks, and when it comes to downstream tasks (such as predicting different variables or super-resolution), fine-tuning of the pre-trained model is required. However, our model can learn different tasks simultaneously through the in-context learning method without the need for further fine-tuning. Even our method exhibits generalization capabilities on unseen OOD tasks without any training or fine-tuning. This is an important attribute of AI-based weather foundation models, as mentioned in existing works[1,2].
> * In the Appendix C section of our paper, we conducted a comparison between our model and ClimaX on the temperature at 2 meters above ground (T2m) prediction task within the ERA5 dataset (WeatherBench). ClimaX fine-tunes the pre-trained model for each lead time. In other words, ClimaX requires N models for weather forecasting at N lead times. In contrast, our universal model can handle all lead time tasks using just a single model.
>
> [1] Nguyen T, Brandstetter J, Kapoor A, et al. ClimaX: A foundation model for weather and climate[J]. arXiv preprint arXiv:2301.10343, 2023.
>
> [2] Chen S, Long G, Jiang J, et al. Foundation models for weather and climate data understanding: A comprehensive survey[J]. arXiv preprint arXiv:2312.03014, 2023.
>
>
> #### W5. There are two clerical errors: (a) "the" should be capitalized. (b) The basic learning rate in line 366 should be 1e-4.
>
> We have revised the description in our main paper, thanks to your suggestions!

---

> > ### Comment · Reviewer_qe7B · 2024-11-24
> > **Good response**
> >
> > Thanks for your detailed response! I will raise my score.

---

> > > ### Author Response · Authors · 2024-11-24
> > > **Thank you**
> > >
> > > Dear Reviewer qe7B,
> > >
> > > Thank you for reading our reply immediately and raising the score. We appreciate your professional suggestion and have incorporated it in our revised version. We're grateful again for your valuable feedback.

---

> > > > ### Author Response · Authors · 2024-12-02
> > > >
> > > > Thank you for your recognition of our work. We have provided General Comments and a Revision Summary, and we appreciate your constructive feedback that has helped make our paper better. We also welcome your participation in further discussions regarding the General Comments.

---

> > > > > ### Comment · Reviewer_qe7B · 2024-12-03
> > > > > **Good General Comments**
> > > > >
> > > > > I have reviewed the general comments and the paper revision. I have also carefully read the other comments and the corresponding feedback. Although this paper may have some minor issues, they do not affect the core contributions. In my view, this paper highlights the versatile capabilities on weather understanding tasks and offers reasonable solutions. I believe the exploration of the versatile capabilities of weather models is insightful and can inspire further research.
> > > > >
> > > > > Overall, there is no studies that explore a generalist model to address such a diverse range of weather understanding tasks. It can even handle medium-term atmospheric forecasting tasks, which is very impressive. Reviewer vbFm also agrees with my perspective. Considering the positive rating from Reviewer vbFm and Broi, and Reviewer vbFm also believes that Reviewer CC3i's demands are unfair, I will further raise my score.

---

### Official Review · Reviewer_CC3i · 2024-11-01

**Soundness:** 1
**Presentation:** 2
**Contribution:** 2
**Rating:** 3
**Confidence:** 3

**Summary:**

The authors propose a generalist foundation model for weather tasks that includes in-context learning that can handle multiple weather tasks.

**Strengths:**

**(S1)**: The problem of creating a more general weather foundation model is important, and it is valuable that the authors are investigating this area.

**(S2)**: Demonstrating in-context learning is also useful for weather foundation models, although I admit I haven’t looked too closely into whether there exists prior work that tackles this.

**Weaknesses:**

**(W1)**: Experimental comparisons are missing. The authors don’t seem to compare WeatherGFM on the same tasks used in the Climax or Aurora papers, and it is unclear why this comparison isn’t made. The main claim is that WeatherGFM is a more general foundation model, and yet the only comparison is made with simple task-specific baselines such as a ViT or UNet. There needs to be compelling evidence that WeatherGFM can outperform/compete with Climax or Aurora. Moreover, only 3 qualitative examples on out-of-distribution (OOD) examples are given in Figure 6. Quantitative evaluations on OOD tasks are quite important for a foundation model.

**(W2)**: Experimental results are weak. For the image translation tasks in Table 2, it is unclear that WeatherGFM can outperform baselines on these datasets. The authors do mention that their goal is not to achieve SoTA performance on each task— this is fine, but since comparison with other SoTA methods is missing, the results on the tasks described are not compelling enough. Moreover, in weather forecasting and super-resolution, the performance of the ViT baseline is nearly the same as WeatherGFM, which suggests only a minute (if any) improvement with the author’s designed modifications and pre-training strategy. Moreover, details on how these baseline ViT or UNets were trained are lacking, which are very crucial to determine if the experiments are fair.

**(W3)**: Lack of significant novelty. While the authors do propose changes to the ViT pre-training setup used in other Weather foundation model setups, these are not significant. The authors’ claims of supporting multi-modality isn’t well supported considering other foundation models already do support inputs from different weather sensors and earth observation (EO) systems. The only difference here seems to be satellite imagery which I don’t think is too different from the setup of the other foundation models, and the authors don’t provide any evidence that the techniques of Climax or Aurora wouldn’t also extend to this additional input. Masked image modelling for EO data and designing patch embeddings for different sensory inputs is an established technique, eg: [1].

**(W4)**: Lack of presentation clarity. I think this paper is tackling an important problem, but does not clearly motivate its solution.
* For example, figure 3 is quite unclear and it is very hard to understand what exactly is going on (in terms of the pre-training strategy or the architectural modifications). The masked/predicted/ground-truth patch disambiguation is not clear to me. It seems that this figure contains multiple subfigures that should clearly be disambiguated and explained.
* The actual pre-training method is not described extensively. Specifically, lines 236-253 need to be clearly explained and expanded, since this seems to be the crux of the paper’s modifications. Instead, too much space is given to explaining the architecture of a ViT (lines 254-280), which is not required in the main text of the paper.
* Qualitative outputs from the model are not compared with qualitative outputs from other baselines. The authors demonstrate the outputs of WeatherGFM but don’t compare these on tasks with other weather foundation models. This is related to (W1).
* Critical details on training and inference are missing. What is the patch size? What kind of compute was used for training? How much compute? Was the dataset combined? Any dataset augmentation? etc. etc. etc. these details are important for reproducibility but are missing from the paper and appendix.

Overall, given the weaknesses describe above, I don't think this paper is ready for acceptance in this conference.

References:
[1] SatMAE: Pre-training Transformers for Temporal and Multi-Spectral Satellite Imagery, NeurIPS 2022.

**Questions:**

See Weaknesses.

Are there ablations on the length of the input prompt sequence and how that affects performance?
Keeping text bold in the introduction paragraphs hampers readability.
Line 159-180: Define what t is and what its range is

---

> ### Author Response · Authors · 2024-11-23
> **Response to Reviewer CC3i (Part 1)**
>
> Dear Reviewer CC3i,
>
> Thank you for your thorough feedback. However, we feel there may have been some misinterpretations of crucial elements of our work, which could influence your assessment. We earnestly hope that our clarifications will prompt you to revisit and reassess our work.
>
> #### W1: Lack of significant novelty.
>
> > 1.1 While the authors do propose changes to the ViT pre-training setup used in other Weather foundation model setups, these are not significant.
>
> We'd like to highlight the **differences between the pre-trained foundation models (Climax, Aurora and SateMAE) and the generalist foundation model (our WeatherGFM).** As stated in Table 1 of our paper, pre-trained foundation models (such as ClimaX, Aurora) all require fine-tuning for downstream tasks.
>
> A pre-trained foundation model typically involves two stages: (1) **pre-training** with mask modeling; and (1) **fine-tuning** for specific tasks. For example, in SatMAE [1], the pre-trained model is fine-tuned for classification and segmentation tasks. In contrast, our approach utilizes a generalist model that **eliminates the need for a fine-tuning process**. Our generalist foundation model adopts the in-context learning paradigm, enabling unified training across multiple tasks simultaneously. It is important to note that the mask modeling within in-context learning is designed for visual question answering, rather than serving as pre-training for later fine-tuning. Consequently, our generalist model does not require any fine-tuning after the training phase.
>
> Reviewer vbFm also recognized the absence of a fine-tuning requirement in our method as a significant strength. Our WeatherGFM can handle different tasks with just visual prompts, eliminating the need for additional fine-tuning. We have incorporated discussions on pre-trained foundation models, including SatMAE [1], SatMAE++ [2], and S2MAE [3], on page 3 of our main paper.
>
> > 1.2 The authors’ claims of supporting multi-modality isn’t well supported considering other foundation models already do support inputs from different weather sensors and earth observation (EO) systems.
>
> As stated in Table 1 of our paper, our WeatherGFM is the first to support single-modal, multi-modal, and temporal modalities of meteorological satellite and radar data for multiple weather undersatanding tasks. To our knowledge, no other work has achieved this. Large remote sensing foundation models (e.g., SatMAE, SatMAE++) tend to conduct image classification and segmentation tasks based on visible and hyperspectral data. Climate foundation models (e.g., ClimaX, Aurora) focus on atmospheric variables forecasting based on reanalysis data. These foundational models do not support the processing of various weather understanding tasks involving meteorological satellite and radar data.
>
> >  1.3 The authors don’t provide any evidence that the techniques of Climax or Aurora wouldn’t also extend to this additional input.
>
> ClimaX and Aurora are capable of accomplishing various downstream tasks through the approach of **fine-tuning**. However, they require separate models for each task, meaning that **for N downstream tasks, N different models must be fine-tuned**. In contrasts, **our generalist model uses a single model to handle all tasks**.
>
> [1] Cong Y, Khanna S, Meng C, et al. Satmae: Pre-training transformers for temporal and multi-spectral satellite imagery[J]. Advances in Neural Information Processing Systems, 2022, 35: 197-211.
>
> [2] Noman M, Naseer M, Cholakkal H, et al. Rethinking transformers pre-training for multi-spectral satellite imagery[C]//Proceedings of the IEEE/CVF Conference on Computer Vision and Pattern Recognition. 2024: 27811-27819.
>
> [3] Li X, Hong D, Chanussot J. S2MAE: A Spatial-Spectral Pretraining Foundation Model for Spectral Remote Sensing Data[C]//Proceedings of the IEEE/CVF Conference on Computer Vision and Pattern Recognition. 2024: 24088-24097.

---

> ### Author Response · Authors · 2024-11-23
> **Response to Reviewer CC3i (Part 2)**
>
> #### W2: Experimental comparisons are missing.
> > 2.1 The authors don’t seem to compare WeatherGFM on the same tasks used in the Climax or Aurora papers, and it is unclear why this comparison isn’t made.
> * Our approach focuses on weather understanding tasks within meteorological satellite and radar data scenarios, whereas ClimaX and Aurora concentrate on atmospheric variable scenarios based on reanalysis data. Reanalysis data scenario focuses on atmospheric variables forecesing, while meteorological satellite and radar observation scenarios involve a more diverse range of tasks, such as weather image translation, spatiotemporal super-resolution and short-term forecasting. Notably, radar short-term forecasting and atmospheric variable forecasting are very different tasks that cannot be directly compared. In fact, our method has proven the capability of generalist models in forecasting tasks for both satellite and radar data.
> * To address your concerns, we have expanded our tasks to include atmospheric variables forecasting. We compared our WeatherGFM with Climax and the operational Integrated Forecasting System (IFS) on WeatherBench. Table 1 shows that our approach significantly outperforms ClimaX in the 120-hour and 168-hour forecast results, even surpassing the IFS method of ECMWF. Our method achieves comparable results to ClimaX at other lead times. It is important to note that Climax fine-tunes the pre-trained model for each lead time, necessitating *N models for weather forecasting at N lead times*. In contrast, our generalist model can **manage all lead time tasks with a single model**, eliminating the need for task-specific fine-tuning.
>
>
> **Note**: Comparison of RMSE and ACC across different lead times for IFS, ClimaX, and ours WeatherGFM on the temperature at 2 meters above ground (T2m) variable. ClimaX views predicting at each lead time as a separate task and fine-tunes a separate model for every individual task. In contrast, our WeatherGFM utilizes a single model to deal with all of these tasks.
> | Lead Time(hr.) | RMSE$\downarrow$ | RMSE$\downarrow$ | RMSE$\downarrow$ | ACC$\uparrow$ | ACC$\uparrow$ | ACC$\uparrow$ |
> |-----------|------|--------|------|------|--------|------|
> |           | IFS  | ClimaX | WeatherGFM | IFS  | ClimaX | WeatherGFM |
> | 6         | **0.97** | 1.11   | 1.08 | **0.99** | 0.98   | 0.98 |
> | 24        | **1.02** | 1.19   | 1.23 | **0.99** | 0.97   | 0.97 |
> | 72        | **1.30** | 1.47   | 1.56| **0.98** | 0.96   | 0.96 |
> | 120       | 1.71 | 1.83   | **1.68** | **0.96** | 0.94   | 0.95 |
> | 168       | 2.23 | 2.17   | **1.76** | 0.93 | 0.91   | **0.94** |
>
> >2.2 Moreover, only 3 qualitative examples on out-of-distribution (OOD) examples are given in Figure 6. Quantitative evaluations on OOD tasks are quite important for a foundation model.
>
> The quatitative results have been added to Appendix H. We conduct generalization tests directly on out-of-distribution (OOD) tasks without any training or fine-tuning of our model. In contrast, the specialized models UNet and ViT are retrained for each task. Quantitative evaluation shows that our method outperforms the specialised ViT model in IR107 Satellite Extrapolation. Although there is a slight performance decrease in Tasks IR107 to IR069 and Temporal SR at 15min compared to ViT models, our WeatherGFM still significantly surpasses the results of the specialized UNet models. **Our experiments demonstrate that our approach maintains generalizability in OOD tasks without requiring any fine-tuning.**
>
> #### Quantitative evaluation on OOD tasks
>
> Note: `*` denotes that **our WeatherGFM has not been trained or fine-tuned** for the tasks listed below and performs generalized inference directly. `#` indicates that UNet and ViT undergo **supervised training** on the corresponding training dataset.
>
> ##### IR107 Satellite Extrapolation
>
> | Metrics | RMSE$\downarrow$  | CSI/-4K$\uparrow$ | CSI/0$\uparrow$ | CSI/2K$\uparrow$ |
> |-------|-------|---------|-------|--------|
> | UNet# | 0.991 | 0.695   | 0.642 | 0.074  |
> | ViT#  | 0.413 | 0.899   | **0.776** | **0.245**  |
> | WeatherGFM* | **0.389** | **0.903**   | 0.774 | 0.244  |
>
> ##### IR107 to IR069
>
> | Metrics | RMSE$\downarrow$  | CSI/-4K$\uparrow$ | CSI/-6K$\uparrow$ |
> |-------|-------|---------|---------|
> | UNet# | 0.942 | 0.642   | 0.910   |
> | ViT#  | **0.212** | **0.958**   | **0.986**   |
> | WeatherGFM* | 0.340 | 0.934   | **0.986**   |
>
> ##### Temporal SR at 15min
>
> | Metrics | RMSE$\downarrow$  | CSI/16$\uparrow$ | CSI/74$\uparrow$ | CSI/133$\uparrow$ | CSI/160$\uparrow$ | CSI/181$\uparrow$ | CSI/219$\uparrow$ |
> |-------|-------|--------|--------|---------|---------|---------|---------|
> | UNet# | 0.676 | 0.211  | 0.627  | 0.428   | 0.351   | 0.262   | 0.083   |
> | ViT#  | 0.218 | 0.838  | 0.761  | 0.598   | 0.525   | 0.445   | 0.190   |
> | WeatherGFM* | 0.272 | 0.814  | 0.703  | 0.507   | 0.419  |0.336   | 0.117   |

---

> ### Author Response · Authors · 2024-11-23
> **Response to Reviewer CC3i (Part 3)**
>
> #### W3: Experimental results are weak.
>
> >In weather forecasting and super-resolution, the performance of the ViT baseline is nearly the same as WeatherGFM, which suggests only a minute (if any) improvement with the author’s designed modifications and pre-training strategy.
>
> * Our WeatherGFM have achieved siginificant performance improvment than task specialized models (e.g., ViT-ST) on most tasks, such as the satellite nowcasting, radar satellite nowcasting, and GOES2Radar translation. It is noteworthy that we employ a single model to handle all tasks without any task-specific fine-tuning.
> * In fact, different tasks present varying levels of learning difficulty. Some simpler tasks are challenging to improve further because they can be easily trained to achieve a relatively high level of performance. This phenomenon can also be witnessed in PromptGIP [1] and GenLV [2]. In our experiments, the satellite radar image super-resolution task is simpler compared to other tasks. For example, the CSI index for the satellite super-resolution task has reached around 0.99. Our model has achieved sufficiently good performance.
>
> [1] Liu Y, Chen X, Ma X, et al. Unifying image processing as visual prompting question answering[J]. arXiv preprint arXiv:2310.10513, 2023.
>
> [2] Chen X, Liu Y, Pu Y, et al. Learning a low-level vision generalist via visual task prompt[C]//Proceedings of the 32nd ACM International Conference on Multimedia. 2024: 2671-2680.
>
> #### W4. Presentation clarity.
> >4.1 Figure 3 is quite unclear.
>
> We have revised Figure 3, adding corresponding descriptions for each module and formula in the training and testing process within the figure 3.
>
> >4.2 The actual pre-training method is not described extensively.
>
> In fact, our method section does not involve any pre-training. To help you understand, we have added detailed descriptions of Mixed-modal mask modeling of Weather in-context learning in page5 of main paper.
>
> >4.3 Qualitative outputs from the model are not compared with qualitative outputs from other baselines.
>
> In our response to Weakness 1 and 2, we provided a comparison with Climax and the quantitative evaluation on OOD tasks.
>
> >4.4 Critical details on training and inference are missing
>
> We have added a detailed description of data construction on page 6 of the main text, and on page 15 of the supplementary materials, we have included descriptions of the network architecture and training details for the baseline UNet, ViT, and our WeatherGFM.
>
> >4.5 Are there ablations on the length of the input prompt sequence and how that affects performance?
>
> During the design of the prompt, its length aligns with that of the model's input query and output target. More precisely, the prompt input ($P_{in}$) and the prompt target ($P_{target}$) are chosen from the training dataset based on the task types associated with the input query ($X_{in}$) and the ground truth ($X_{target}$). In other words, for a particular task, the length of the prompt remains fixed. We have performed an ablation study on the approach of selecting the prompt in section 4.4 of main paper (page. 9). The detailed information can be found in Appendix F.
>
> >4.6 Keeping text bold in the introduction paragraphs hampers readability. Line 159-180: Define what t is and what its range is.
>
> We have revised the description in our main paper, thanks to your suggestions.

---

> > ### Comment · Reviewer_CC3i · 2024-11-29
> > **Response to Author Rebuttal**
> >
> > Thank you for presenting a detailed rebuttal. I appreciate the additional experiments, but there are still many questions that I think are unanswered.
> >
> > Thank you for an initial comparison with Climax. A few questions on this experiment:
> > * Why was only T2m chosen for the comparison? And why not for the entire range of leadtimes shown in the Climax paper?
> > * "N models for weather forecasting at N lead times"- I see that the Climax paper also suggests a method to avoid fine-tuning a model for each leadtime (4.6.1) but that it underperforms for T2 specifically. Since WeatherGFM is proposed as a foundation model for weather tasks, it is important to have a comprehensive comparison with Climax.
> > * Comparison with Aurora is still missing.
> > * In section C, was a new Weather GFM trained for the Era5 comparison? Or was the original model used (from the previous experiments)? Or was the data mixed with the previous tasks as an additional task? These questions are important to answer to claim generality.
> >
> > Some other details:
> > > In fact, our method section does not involve any pre-training
> >
> > This confuses me. Isn't the goal of this work to present a foundation model? Are you starting from an initialization for the ViT from some other pre-trained source? Or are you classifying the masked-image training as "fine-tuning"?
> >
> > > prompt length aligns with that of the model's input query and output target
> >
> > But how are the prompts selected during training? Is the prompt always fixed during training, or randomized? What is the impact of this selection during training? Section 4.4 and Table 3 seems to be an inference-time study. Also I see this line in the appendix "In our main paper, we detail a thorough process of prompt selection". Where is this described exactly? And what are the 20 unique prompts? And what is idx0, idx9 etc. in Table 8? My question still remains though-- how does increasing or decreasing the number of in-context examples during training or testing impact performance.
> >
> > Other thoughts--
> > * In equation (4), what is T?
> > * In section 4.1, specifying which datasets correspond to which of the 3 prompt formats from figure 2 would be valuable. Also, how many in-context examples are given for each of these datasets. This information on how prompts were selected should not be left to Appendix F, and lines 958-962 are not clear to me. In general, the description of data formats in section 4 is not clear-- the key elements that should be specified are exactly what the input and outputs are (i.e. size, what they correspond to), what the prompt formats are, and how the prompts were selected (eg: are they fixed for each query during inference). I found myself having to constantly jump between section 4 and the appendix to piece together relevant details on the data for each task, and this significantly hampers presentation.
> > * For super-resolution tasks, there are many super-resolution specific models that are not ViT or UNet baselines (StableDiffusion is just 1 example, there are many others). Why weren't these used for comparison?

---

> > > ### Author Response · Authors · 2024-11-29
> > > **Response to Reviewer CC3i (Part 1)**
> > >
> > > Dear Reviewer CC3i,
> > >
> > > We really appreciate your response to our rebuttal and thank you for informing us about the aspects that were unclear to you. We are happy to provide further clarificatin to your questions.
> > >
> > >
> > > #### Questions on experiment.
> > >
> > > > 1.1 Why was only T2m chosen for the comparison? And why not for the entire range of leadtimes shown in the Climax paper?
> > >
> > >
> > > Our weathGFM model emphasizes a broader range of weather understanding tasks beyond just atmospheric forecasting. As a result, we selected a subset of variables to assess our model's general capability in atmospheric forecasting. To address your concerns further, we conducted additional experiments focusing on the zonal wind speed at 10 meters above the ground (U10). The quantitative results, presented in the table, demonstrate that our method outperforms ClimaX in nearly all lead times.
> > >
> > > We utilize the commonly used medium-range forecasting setting, which spans from 6 hours to 7 days. This setting is often employed by medium-range forecasting models such as GraphCast, Pangu-Weather, and FourCastNet.
> > >
> > >
> > > **Note**: This is a comparison of RMSE and ACC across different lead times for IFS, Aurora, ClimaX, and our WeatherGFM on the **U10** variable. ClimaX treats predicting at each lead time as a separate task and *fine-tunes a separate model for every individual task*. In contrast, our WeatherGFM utilizes *a single model to handle all these tasks*. **Aurora** is a forecasting foundation model that is **trained using higher resolution atmospheric data from ERA5 (0.25°)**, making **its results not directly comparable**.
> > >
> > > | Lead Time(hr.) | RMSE$\downarrow$ | RMSE$\downarrow$ | RMSE$\downarrow$ | RMSE$\downarrow$| ACC$\uparrow$ | ACC$\uparrow$ | ACC$\uparrow$ |
> > > |-----------|------|-----|---|------|------|--------|------|
> > > |           | IFS (1.40625°)  | Aurora (0.25°) |ClimaX (1.40625°) |WeatherGFM (1.40625°) | IFS (1.40625°)  | ClimaX (1.40625°) | WeatherGFM (1.40625°) |
> > > | 6         | **0.79** | *0.69* |1.04   |1.12 |**0.98** | 0.97   | 0.97 |
> > > | 24        | **1.11** |  *0.97* |1.31   |1.26 |**0.97** | 0.95   | 0.95 |
> > > | 72        | **1.92** |  *1.56* |2.02   |1.99| **0.89** | 0.87   | 0.88 |
> > > | 120       | 2.89|  *2.27* |2.79 |**2.61** |0.76 | 0.74   | **0.79** |
> > > | 168       | 3.81 |  *2.98* |3.35  |**3.11** |0.58 | 0.59   | **0.65** |
> > >
> > > **Note**: This ia a comparison of RMSE and ACC across different lead times for IFS, Aurora, ClimaX, and our WeatherGFM on the **T2m** variable.
> > > | Lead Time(hr.) | RMSE$\downarrow$ | RMSE$\downarrow$| RMSE$\downarrow$ | RMSE$\downarrow$ | ACC$\uparrow$ | ACC$\uparrow$ | ACC$\uparrow$ |
> > > |-----------|------|----|----|------|------|--------|------|
> > > |           | IFS (1.40625°)  | Aurora (0.25°) |ClimaX (1.40625°) |WeatherGFM (1.40625°) | IFS (1.40625°)  | ClimaX (1.40625°) | WeatherGFM (1.40625°) |
> > > | 6         | **0.97** |*0.53* |1.11   | 1.08 | **0.99** | 0.98   | 0.98 |
> > > | 24        | **1.02** |*0.68* |1.19   | 1.23 | **0.99** | 0.97   | 0.97 |
> > > | 72        | **1.30** | *0.96*|1.47   | 1.56| **0.98** | 0.96   | 0.96 |
> > > | 120       | 1.71 |*1.32* |1.83   | **1.68** | **0.96** | 0.94   | 0.95 |
> > > | 168       | 2.23 | *1.73*|2.17   | **1.76** | 0.93 | 0.91   | **0.94** |

---

> > > ### Author Response · Authors · 2024-11-29
> > > **Response to Reviewer CC3i (Part 2)**
> > >
> > > > 1.2 "N models for weather forecasting at N lead times"- I see that the Climax paper also suggests a method to avoid fine-tuning a model for each leadtime (4.6.1) but that it underperforms for T2 specifically. Since WeatherGFM is proposed as a foundation model for weather tasks, it is important to have a comprehensive comparison with Climax.
> > >
> > > We compare our results to ClimaX's best performance, which involves fine-tuning separate models for each of the N lead times. Our results have already surpassed ClimaX's best performance on the U10 and T2m variables. ClimaX is designed to address a variety of **atmospheric** tasks, such as atmospheric forecasting and super-resolution, but it requires fine-tuning for each specific task. In contrast, our WeatherGFM is designed for a broader range of **meteorological satellite and weather radar understanding** tasks, including forecasting, spatio-temporal super-resolution, weather image translation, and post-processing, all without the need for additional fine-tuning. Consequently, our generalist model not only handles typical atmospheric variable forecasting tasks with a single model but also performs 10 different meteorological satellite and radar tasks. As shown in the table below, atmospheric forecasting tasks encompass numerous sub-tasks, with each variable and level representing a distinct sub-task. Our comparison on the two commonly used variables demonstrates that our model can effectively extend to forecasting tasks in the ERA5 dataset.
> > >
> > >
> > > | Type         | Variable name       | Abbrev. | ECMWF ID | Levels                      |
> > > |--------------|---------------------|---------|----------|-----------------------------|
> > > | Static       | 2 metre temperature | T2m     | 167      | Single                      |
> > > | Static       | 10 metre U wind component | U10    | 165      | Single                      |
> > > | Static       | 10 metre V wind component | V10    | 166      | Single                      |
> > > | Atmospheric  | Geopotential        | Z       | 129      | 50, 250, 500, 600, 700, 850, 925 |
> > > | Atmospheric  | U wind component    | U       | 131      | 50, 250, 500, 600, 700, 850, 925 |
> > > | Atmospheric  | V wind component    | V       | 132      | 50, 250, 500, 600, 700, 850, 925 |
> > > | Atmospheric  | Temperature         | T       | 130      | 50, 250, 500, 600, 700, 850, 925 |
> > > | Atmospheric  | Specific humidity   | Q       | 133      | 50, 250, 500, 600, 700, 850, 925 |
> > > | Atmospheric  | Relative humidity   | R       | 157      | 50, 250, 500, 600, 700, 850, 925 |
> > > > 1.3 Comparison with Aurora is still missing.
> > >
> > > To address your concern, we have included a comparison with the Aurora model. The Aurora model is a forecasting foundation model specifically designed for tasks like atmospheric forecasting and air quality forecasting. In contrast, the ClimaX model is versatile, handling a variety of atmospheric tasks, including atmospheric forecasting and super-resolution. Our WeatherGFM model is designed for an even broader range of meteorological tasks, such as meteorological satellite and radar forecasting, spatio-temporal super-resolution, weather image translation, post-processing, and over ten other tasks.
> > >
> > > In the newly added atmospheric forecasting experiments, our generalist model has outperformed ClimaX on multiple variables. However, it is not crucial for either ClimaX or WeatherGFM to surpass the Aurora model, as Aurora is specifically optimized for forecasting tasks using large-scale, higher-resolution data and a greater number of model parameters.
> > >
> > >
> > > **Data Scale**: Aurora leverages a substantial 1.2 PB of data across ten datasets, including 40 years of ERA5 data, 65 years of CMCC data, and 65 years of ECMWF-IFS-HR data. In contrast, we utilized the WeatherBench dataset, consistent with ClimaX, which is 9 TB in size—significantly smaller than Aurora's 1.2 PB.
> > >
> > > **Model Scale**: Aurora's model comprises 1.3 billion parameters, considerably larger than our model, which has 300 million parameters.
> > >
> > > **Resolution**: Atmospheric forecasting commonly uses resolutions such as 5.625°, 1.40625°, 0.25°, and 0.1°. Aurora, focusing on forecasting tasks, employs resolutions of 0.25° and 0.1°, which often produce better quantitative results than 1.4°. For comparison with ClimaX, we selected the 1.4° setting for our atmospheric forecasting experiments. Our generalist model has outperformed ClimaX on multiple variables in these experiments.

---

> > > ### Author Response · Authors · 2024-11-29
> > > **Response to Reviewer CC3i (Part 3)**
> > >
> > > > 1.4 In section C, was a new WeatherGFM trained for the Era5 comparison? Or was the original model used (from the previous experiments)? Or was the data mixed with the previous tasks as an additional task? These questions are important to answer to claim generality.
> > >
> > > The WeatherGFM model used for the ERA5 comparison is further trained on the original model by incorporating the ERA5 data, mixed with the previous tasks, as an additional task for continual training.
> > >
> > > #### Other Details.
> > > > 2.1 Isn't the goal of this work to present a foundation model? Are you starting from an initialization for the ViT from some other pre-trained source? Or are you classifying the masked-image training as "fine-tuning"?
> > >
> > > We apologize for the confusion. Our foundation model is trained from scratch. What we meant to convey is that after training, it can be directly applied to handle different tasks without requiring fine-tuning.
> > >
> > > > 2.2 How are the prompts selected during training? Is the prompt always fixed during training, or randomized? What is the impact of this selection during training? Section 4.4 and Table 3 seems to be an inference-time study. Also I see this line in the appendix "In our main paper, we detail a thorough process of prompt selection". Where is this described exactly? And what are the 20 unique prompts? And what is idx0, idx9 etc. in Table 8? My question still remains though-- how does increasing or decreasing the number of in-context examples during training or testing impact performance.
> > >
> > >
> > > Following PromptGIP [1] and GenLV [2], we randomly select prompts for each training sample from the training set of the same task, rather than using fixed prompts. During both training and inference, in-context prompts are randomly selected from all samples in the training set. The statement in the appendix, "In our main paper, we detail a thorough processin of prompt selection," refers to lines 478-483 of our main paper. This section is part of an ablation study aimed at verifying the impact of different prompts during inference. Specifically, we randomly selected 20 fixed prompts from the training set to calculate the standard deviation across different prompts, using idx0 to idx19 to represent these 20 prompts. The prompt input ($P _ {in}$) and the prompt target ($P _ {target}$) are chosen from the training dataset based on the task types associated with the input query ($X _ {in}$) and the ground truth ($X _ {target}$). In summary, during both training and testing, prompts are randomly selected from the training set, and the entire training set is used as in-context examples. Therefore, there is no need to adjust the number of prompts. Our ablation studies in Section 4.4 and Appendix F demonstrate that different prompts have minimal impact on performance.
> > >
> > > >2.3 In equation (4), what is T?
> > >
> > > We apologize for the typographical error. The correct version of equation (4) should be:
> > >
> > > $$
> > > \mathrm{L}^{total} _ {\theta} = \mathrm{L} _ 2(P^{'} _ {target},P _ {target}) + \mathrm{L} _ 2(X^{'} _ {target},X _ {target}).
> > > $$
> > >
> > > > In section 4.1, specifying which datasets correspond to which of the 3 prompt formats from figure 2 would be valuable. Also, how many in-context examples are given for each of these datasets. This information on how prompts were selected should not be left to Appendix F, and lines 958-962 are not clear to me. In general, the description of data formats in section 4 is not clear-- the key elements that should be specified are exactly what the input and outputs are (i.e. size, what they correspond to), what the prompt formats are, and how the prompts were selected (eg: are they fixed for each query during inference). I found myself having to constantly jump between section 4 and the appendix to piece together relevant details on the data for each task, and this significantly hampers presentation.
> > >
> > >
> > >
> > > Thank you for the valuable suggestion. During both the training and testing processes, in-context examples are randomly selected from the training set. Therefore, the number of in-context examples matches the size of the training set. In lines 958-962, we describe the process of selecting different prompts for the ablation studies conducted during inference. The term “High prompt” refers to prompts selected from a high-quality prompt base derived from radar data. Specifically, within the training set, we grouped every 100 events and selected images with relatively high extreme values from each group, focusing on radar images with the most pixels exceeding the threshold of 50. Additionally, we identified images most similar to the input images as the "Searched prompt" using the Root Mean Square Error (RMSE) metric. To enhance clarity regarding data format, we have organized the model's input, output, prompt input, prompt output, prompt formats, and prompt selection into the following table. For every task, the images' size are 256X256. This should improve the overall presentation of our work.

---

> > > ### Author Response · Authors · 2024-11-29
> > > **Response to Reviewer CC3i (Part 4)**
> > >
> > > | Task | Dataset| Prompt Formate| Prompt Input | Prompt Output |Input|Output|Prompt Selection|
> > > | -------- | --|---|--- | -------- |-------- |-------- |-------- |
> > > |Radar Spatial SR   | Sevir| Single Modal  |  VIL LR|VIL HR|VIL LR|VIL HR|Randomized|
> > > |Satellite Spatial SR   | Sevir| Single Modal  |  IR-069 LR|IR-069 HR|IR-069 LR|IR-069 HR|Randomized|
> > > |Radar Temporal SR   | Sevir| Single Modal  |  VIL (0,60min)|VIL 30min|VIL (0,60min)|VIL 30min|Randomized|
> > > |Deblur   | Sevir| Single Modal  |  VIL (Earthformer)|VIL (Ground Truth)|VIL (Earthformer)|VIL (Ground Truth)|Randomized|
> > > |GEOS-IR2Radar   | Sevir     | Cross Modal  |  IR-069,IR-107|VIL|IR-069,IR-107|VIL|Randomized|
> > > |GEOS-IR2GEOS-IR   | Sevir  | Cross Modal  |  IR-069|IR-107|IR-069|IR-107|Randomized|
> > > |GEOS-IR2GEOS-Vis   | Sevir  | Cross Modal  |  IR-069|VIS|IR-069|VIS|Randomized|
> > > |GEOS2POES-NO2   | POMINO-TROPOMI, GEOS-CF  | Cross Modal  |   GEMS, GEOS-CF | TROPOMI| GEMS, GEOS-CF | TROPOMI|Randomized|
> > > |Satellite extrapolation    | Sevir| Time-series Modal  |  IR-069  (0,30,60,90min)|IR-069  (120,180min) |IR-069  (0,30,60,90min)|IR-069  (120,180min)|Randomized|
> > > |Radar extrapolation    | Sevir | Time-series Modal  |  VIL  (0,30,60,90min)|VIL  (120,180min)|VIL  (0,30,60,90min)|VIL  (120,180min)|Randomized|
> > >
> > >
> > > > 2.4 For super-resolution tasks, there are many super-resolution specific models that are not ViT or UNet baselines (StableDiffusion is just 1 example, there are many others). Why weren't these used for comparison?
> > >
> > > Super-resolution models designed for natural images, which are typically trained on three-channel RGB data, often cannot be directly applied to atmospheric, satellite, and radar images. This is because these types of data frequently contain many more channels; for example, ClimaX's atmospheric data includes 48 variables (channels). As a result, existing super-resolution models are not directly applicable to these datasets. In ClimaX, atmospheric super-resolution tasks are compared using models like ResNet, UNet, and ViT. Following the ClimaX approach, we utilize UNet and ViT to conduct fair comparisons for super-resolution tasks involving satellite and radar images.

---

> > > ### Author Response · Authors · 2024-12-01
> > > **Follow-Up on Rebuttal Clarifications**
> > >
> > > Dear Reviewer CC3i,
> > >
> > > We hope this message finds you well. We apologize for any inconvenience caused by reaching out over the weekend. We greatly appreciate your response to our rebuttal and thank you for highlighting the aspects that were unclear.
> > >
> > > We have provided further clarification to address your questions. As **the rebuttal discussion period ends in two days**, we would be grateful for your feedback on whether our responses have adequately addressed your concerns. We are ready to answer any further questions you may have.
> > >
> > > Thank you for your valuable time and effort!
> > >
> > > Best regards,
> > >
> > > The Authors

---

> > > ### Author Response · Authors · 2024-12-02
> > > **Gentle Reminder: Rebuttal Discussion Deadline Approaching**
> > >
> > > Dear Reviewer CC3i,
> > >
> > > We hope this message finds you well. With less than 30 hours remaining until the discussion period concludes, we would greatly appreciate your feedback on whether our recent clarifications have adequately addressed your concerns.
> > > Your insights are invaluable to us, and we are eager to ensure that all your questions are fully resolved. Please let us know if there is anything further we can provide to assist in your review.
> > >
> > > Thank you once again for your time and effort.
> > >
> > > Best regards,
> > >
> > > The Authors

---

> ### Author Response · Authors · 2024-11-27
> **Kindly Requesting for Review of Our Response**
>
> Dear Reviewer CC3i,
>
> **We are eager to ensure that we have adequately addressed your concerns** and are prepared to offer further clarifications or address any additional questions you may have.
>
> Should you find that our revisions have satisfactorily addressed your previous concerns, **we would be most grateful if you would reconsider the evaluation of our paper with a view to enhancing its standing**.
>
> We would like to express our heartfelt gratitude for the time and effort you have dedicated to reviewing our work. **It has been a pleasure to engage with you throughout this process**.
>
>
> Best regards,
>
> Authors

---

> ### Author Response · Authors · 2024-11-28
> **Gentle Reminder**
>
> Dear Reviewer CC3i,
>
> We hope this message finds you well. As we approach the conclusion of the rebuttal discussion, we would be grateful if you could spare some time to go through our response. We are eager to know if we have adequately addressed your concerns, and we’d be more than happy to provide further clarifications or answer any additional questions you may have.
>
> Once again, thank you for your time and dedication to this process.
>
> Best Regards,
>
> The Authors

---

> ### Comment · Reviewer_qe7B · 2024-12-03
> **Different opinions to Reviewer CC3i**
>
> After reading general comments, I would like to offer a different opinion because I am very familiar with weather forecasting.
>
> The proposed generalist model primarily addresses many interrelated tasks within satellite and radar data scenarios, which are significantly different from medium-range forecasting tasks. In my experience, medium-range forecasting tasks involve hundreds of variable levels, with a complexity of optimization that far exceeds weather understanding tasks from satellite and radar. Nonetheless, the generalist model has even outperformed ClimaX in some representative variables. This finding is deeply impressive. Therefore, in my view, the current experimental results sufficiently validate the capabilities of atmospheric forecasting.
>
> Additionally, following ClimaX's approach and directly employing UNet and ViT to construct a baseline is reasonable. This common comparative method is often used to verify model performance across different meteorological scenarios. Thus, as the first attempt to explore a generalist model, I believe the current experiments are sufficiently comprehensive, and further investigation may be reserved for future studies. In my opinion, being too harsh is not helpful to the community.

---

### Official Review · Reviewer_Broi · 2024-11-05

**Soundness:** 3
**Presentation:** 2
**Contribution:** 3
**Rating:** 6
**Confidence:** 3

**Summary:**

This paper presented a weather generalist foundation model based on in-context learning, which unified a wide range of weather understanding tasks, like weather forecasting, wealth image translation, wealth super-resolution, deblur, etc.
It first defines the visual prompt template and then does in-context learning to achieve such a unified model.
The experiments demonstrate that this unified model achieved comparable results with specialist models, e.g., ViT and Unet.

**Strengths:**

1. A good contribution of this work is to apply in-context learning to weather-understanding tasks for building a unified model and confirming its effectiveness.
2. The proposed model demonstrates comparable results with specialist models on ten weather understanding tasks,
3. Mixed-modal mask modeling is interesting.

**Weaknesses:**

1. The motivation of the training objective in Figure 3 is unclear. The eq.5 is mismatched with Figure 3. There should be more details about the training objective. X_T in eq.5 lacks a definition for your universal representation.
2. Details missing. For example, the patch embedding layer is task-specific. Does this mean that there is a hard code to select which patch embedding layer is used to handle different tasks or data modalities? If so, can this model be claimed as a unified model?
3. The motivation of such a unified model is actually unclear to real-world applications. There is no obvious improvement over the conventional specialist model. In addition, there are not many combinations of tasks here. What is the essential advance of the proposed model compared with per-task specialists?

**Questions:**

What is the motivation for designing Mixed-modal mask modeling? It seems like there should be some references that motivate this design.

---

> ### Author Response · Authors · 2024-11-24
> **Response to Reviewer Broi (Part 1)**
>
> Dear Reviewer Broi,
>
> Thank you for your thoughtful and positive feedback. We have addressed the concerns raised in your comments below and hope that our clarifications will further enhance your confidence in our work.
>
> #### W1: The training objective in Figure 3 is unclear.
> > What is the motivation for designing Mixed-modal mask modeling? It seems like there should be some references that motivate this design.
>
> Thank you for your suggestions. we have added motivation for designing Mixed-modal mask modeling and references in the main text, please refer to page 5. Inspired by the concept of Visual Question Answering [1,2,3], we introduce mixed-modality masking on various weather modalities for visual question-and-answer modeling in weather understanding tasks. This process can be formulated as follows:
> $$ P' _ {\text{target}}, X' _ {\text{target}} = F _ {\tau} (P _ {\text{in}}, M(P _ {\text{target}}), X _ {\text{in}}, M(X _ {\text{target}}); \theta); $$
> where we randomly conduct mask operation $M$ on the prompt target $P _ {target}$ as well as the ground truth $X _ {target}$ according to the mask ratio. Meanwhile, the prompt input $P_{in}$ and the input query $X_{in}$ will be retained entirely.  $P^{'} _ {target}$ and $X^{'} _ {target}$ represent the predicted target output of model $F _ {\tau}$.
> The optimization objectives are as follows:
> $$
> \mathrm{L}^{total} _ {\theta} = \mathrm{L} _ 2(P^{'} _ {target},P _ {target}) + \mathrm{L} _ 2(T^{'} _ {target},T _ {target}).
> $$
> where we use MSE (mean square error) loss $\mathrm{L _ 2}$ to train the weather generalist foundation model. In the inference stage, we keep the $P _ {in}$, $P _ {out}$, and $X _ {in}$ intact while the target image is fully masked. This target full masking strategy allows generalist foundation models to generate the corresponding target through a visual question-and-answer format.
>
> [1] Wang X, Wang W, Cao Y, et al. Images speak in images: A generalist painter for in-context visual learning[C]//Proceedings of the IEEE/CVF Conference on Computer Vision and Pattern Recognition. 2023: 6830-6839.
> [2] Liu Y, Chen X, Ma X, et al. Unifying image processing as visual prompting question answering[J]. arXiv preprint arXiv:2310.10513, 2023.
> [3] Chen X, Liu Y, Pu Y, et al. Learning a low-level vision generalist via visual task prompt[C]//Proceedings of the 32nd ACM International Conference on Multimedia. 2024: 2671-2680.
>
> #### W2: The patch embedding layer is task-specific.
>
> > The patch embedding layer is task-specific. Does this mean that there is a hard code to select which patch embedding layer is used to handle different tasks or data modalities? If so, can this model be claimed as a unified model?
>
> Our patch-embedding layer provides a degree of generalizability due to its modality-specific design. Furthermore, it can effectively accommodate new modalities.
> * We currently use three types of modality-specific patch-embedding layers that can handle 10 tasks. If a new task fits within these three modalities, there's no need to add a new embedding layer. For instance, the carbon dioxide image translation task using a new dataset can directly utilize the existing embedding layer without requiring any modifications.
> * If a new modality emerges, our framework can adapt by adding a new modality-specific patch-embedding layer to accommodate tasks related to this new modality. Just as in the Appendix C section of the paper, we conducted a comparison between our model and ClimaX on the task of predicting the temperature at 2 meters above ground (T2m) within the WeatherBench. In this task, the input consists of 48 ECMWF (European Centre for Medium-Range Weather Forecasts) variables, which are significantly different from the inputs of our original tasks. For this new task, we introduced a new patch embedding and adopted the approach of continuous learning to train for it, and ultimately achieved competitive results when compared to specialized models.
> * Unified AI agents in the general domain, such as NeXT-GPT [1] and UNIFIED-IO [2], will also utilize different task-specific encoders for different modalities when handling inputs of various modalities. This implies that when dealing with newly introduced modalities, these models still need to adopt new encoders. The core of WeatherGFM is to construct a generalist model that can handle multiple tasks with just a single model. Meanwhile, when encountering tasks involving new modalities, WeatherGFM is scalable. As this is the first work on a weather generalist foundation model, there is potential for future advancements to enhance our model's adaptability across different modalities.

---

> ### Author Response · Authors · 2024-11-24
> **Response to Reviewer Broi (Part 2)**
>
> **Note**: Comparison of RMSE and ACC across different lead times for IFS, ClimaX, and ours WeatherGFM on the temperature at 2 meters above ground (T2m) variable prediction task. ClimaX views predicting at each lead time as a separate task and *fine-tunes a separate model for every individual task*. In contrast, **our WeatherGFM utilizes a single model to handle all of these tasks**.
> | Lead Time(hr.) | RMSE$\downarrow$ | RMSE$\downarrow$ | RMSE$\downarrow$ | ACC$\uparrow$ | ACC$\uparrow$ | ACC$\uparrow$ |
> |-----------|------|--------|------|------|--------|------|
> |           | IFS  | ClimaX | WeatherGFM | IFS  | ClimaX | WeatherGFM |
> | 6         | **0.97** | 1.11   | 1.08 | **0.99** | 0.98   | 0.98 |
> | 24        | **1.02** | 1.19   | 1.23 | **0.99** | 0.97   | 0.97 |
> | 72        | **1.30** | 1.47   | 1.56| **0.98** | 0.96   | 0.96 |
> | 120       | 1.71 | 1.83   | **1.68** | **0.96** | 0.94   | 0.95 |
> | 168       | 2.23 | 2.17   | **1.76** | 0.93 | 0.91   | **0.94** |
>
>
> [1] Wu S, Fei H, Qu L, et al. Next-gpt: Any-to-any multimodal llm[J]. arXiv preprint arXiv:2309.05519, 2023.
>
> [2] Lu J, Clark C, Zellers R, et al. Unified-io: A unified model for vision, language, and multi-modal tasks[C]//The Eleventh International Conference on Learning Representations. 2022.
>
> #### W3: The motivation on real-world application.
>
> >3.1 There are not many combinations of tasks here.
>
> * In practical application scenarios, regional weather forecasting often requires both weather forecasting and super-resolution tasks to achieve more accurate forecasting results. Weather monitoring scenarios necessitate a variety of weather observation data for analysis, typically requiring models to synthesize missing data, such as generating missing weather radar data, synthesizing missing shortwave infrared and visible light data at night. We can flexibly handle multiple tasks with just single model, eliminating the need to embed multiple models in practical application scenarios.
> * In addition, existing work[1,2,3] has mentioned that meteorological system modeling involves a variety of different tasks that constantly interact in intricate ways.
>
> >3.2 There is no obvious improvement over the conventional specialist model.
> * Our WeatherGFM have achieved siginificant performance improvment than specialized models on most tasks, such as the satellite nowcasting, radar satellite nowcasting, GOES2Radar translation and etc. It is noteworthy that we employ a single model to handle 10 tasks instead of using 10 specialised models for 10 tasks.
> * In fact, different tasks have varying levels of learning difficulty. Some simple tasks are hard to improve further because they are easier to learn to a high performance level. This phenomenon can also be observed in PromptGIP and GenLV. In our experiments, the GOES-IR2GOES-IR image translation task is simpler compared to other tasks. For example, the CSI index for the GOES-IR2GOES-IR task has reached around 0.99. Our model has achieved sufficiently good performance.
>
> >3.3 What is the essential advance of the proposed model compared with per-task specialists?
> * The most critical advancement of our method over specialized models is its versatility and generalization ability. If there are 10 tasks to be processed, specialized models require training 10 separate models while our generalist model only needs a single model to handle all 10 tasks. Specialized model data-driven models struggle to generalize to new tasks, while our model shows versatility across multiple tasks and generalization ability on unseen tasks without any fine-tuning. This is an important attribute of AI-based weather foundation models, as mentioned in existing studies[1,3].
> * In the Appendix C section of our paper, we conducted a comparison between our model and ClimaX[3] on the temperature at 2 meters above ground (T2m) prediction task within the ERA5 dataset (WeatherBench). ClimaX fine-tunes the pre-trained model for each lead time. In other words, ClimaX requires N models for weather forecasting at N lead times. In contrast, our universal model can handle all lead time tasks using just a single model.
>
> [1] Chen S, Long G, Jiang J, et al. Foundation models for weather and climate data understanding: A comprehensive survey[J]. arXiv preprint arXiv:2312.03014, 2023.
>
> [2] Veillette M, Samsi S, Mattioli C. Sevir: A storm event imagery dataset for deep learning applications in radar and satellite meteorology[J]. Advances in Neural Information Processing Systems, 2020, 33: 22009-22019.
>
> [3] Nguyen T, Brandstetter J, Kapoor A, et al. ClimaX: A foundation model for weather and climate[J]. arXiv preprint arXiv:2301.10343, 2023.

---

> ### Author Response · Authors · 2024-11-27
> **Kindly Requesting for Review of Our Response**
>
> Dear Reviewer Broi,
>
> **We are eager to ensure that we have adequately addressed your concerns** and are prepared to offer further clarifications or address any additional questions you may have.
>
> Should you find that our revisions have satisfactorily addressed your previous concerns, **we would be most grateful if you would reconsider the evaluation of our paper with a view to enhancing its standing**.
>
> We would like to express our heartfelt gratitude for the time and effort you have dedicated to reviewing our work. **It has been a pleasure to engage with you throughout this process**.
>
>
> Best regards,
>
> Authors

---

> ### Author Response · Authors · 2024-11-28
> **Gentle Reminder**
>
> Dear Reviewer Broi,
>
> We hope this message finds you well. As we are nearing the end of rebuttal discussion, we would greatly appreciate it if you could find some time in your busy schedule to review our response and let us know if you have any further questions or require additional clarification on any points.
>
> We sincerely appreciate your valuable time and effort.
>
> Best Regards,
>
> The Authors

---

> ### Author Response · Authors · 2024-11-30
> **Kind Reminder**
>
> Dear Reviewer Broi,
>
> We hope this message finds you well. We apologize for any disruption caused by contacting you over the weekend. Thank you for your initial positive rating. As the rebuttal period comes to a close, we would like to confirm whether our rebuttal has resolved your concerns and further enhanced your confidence in our work. We are more than happy to provide further clarifications or answer any additional questions you may have.
>
> Once again, thank you for your time and dedication to this process.
>
> Best regards,
>
> The Authors

---

> ### Author Response · Authors · 2024-12-03
> **Thank you**
>
> Dear reviewer Broi,
>
> We sincerely appreciate your response and are pleased to know that we have largely addressed your concerns. Thank you once again for your valuable feedback, which has significantly contributed to enhancing the quality of our paper.
>
> Best regards,
>
> Authors

---

### Official Review · Reviewer_vbFm · 2024-11-10

**Soundness:** 3
**Presentation:** 2
**Contribution:** 3
**Rating:** 6
**Confidence:** 4

**Summary:**

This paper introduces WeatherGFM, a weather foundation model inspired by in-context learning approaches used in visual and language models. WeatherGFM aims to address multiple weather-related tasks within a unified framework by converting diverse tasks with various data types into general weather prompts. The model employs a masked modeling approach for training, while fully masking the targets during inference. Experimental results on SEVIR dataset show that WeatherGFM can handle up to ten different weather tasks, including forecasting, super-resolution, image translation, and post-processing. The model also demonstrates its generalization capabilities on unseen tasks.

**Strengths:**

- The proposed weather prompt design effectively handles a wide range of weather-related tasks.
- By utilizing an in-context learning approach, WeatherGFM can generalize to unseen tasks without requiring fine-tuning.
- WeatherGFM demonstrates its scalability across different model sizes.
- The model successfully handles ten various tasks on the SEVIR dataset using a single, unified framework.

**Weaknesses:**

- In Section 3.2, it appears that WeatherGFM uses task-specific patch embedding layers, which may limit the framework's generalization ability. For instance, with the current design, WeatherGFM may struggle to generalize to new, unseen tasks that require novel sensor channels.
- The impact of increasing dataset size is unclear in Figure 7. Here, the authors compare ViT-ST with WeatherGFM-Base to illustrate the scalability of WeatherGFM. However, since ViT-ST is trained on a single task while WeatherGFM is trained on multiple tasks with more data and parameters, it’s unclear whether the performance gains are due to dataset scalability or multi-task training or larger number of parameters. Additionally, on radar spatial super-resolution (SR), scaling the dataset size or using multi-task training appears to degrade performance. A more detailed analysis and discussion on the effects of dataset scalability would be valuable.
- It would be beneficial to compare WeatherGFM’s performance against other climate foundation models (e.g., Aurora, ClimaX) using the ERA5 dataset. One of WeatherGFM’s key strengths is its ability to unify multi-task, multi-modal datasets. Thus, similar to Aurora, pretraining WeatherGFM on a combination of ERA5 and other heterogeneous datasets could potentially enhance its performance. It would be impressive if WeatherGFM could achieve comparable results to ClimaX or Aurora without fine-tuning, especially by leveraging a larger dataset and multi-task training.

**Questions:**

- Why does WeatherGFM show weaker performance in weather image translation compared to other tasks?
- In Table 1, there is a misuse of double lines for "GOES2Radar, CSI/219."

---

> ### Author Response · Authors · 2024-11-23
> **Response to Reviewer vbFm (Part 1)**
>
> Dear Reviewer vbFm,
>
> We appreciate your valuable feedback and would like to address the concerns you've raised. We hope the clarifications provided below will resolve your concerns and enhance your evaluation of our work.
>
> #### W1: Framework's generalization ability.
>
> > It appears that WeatherGFM uses task-specific patch embedding layers, which may limit the framework's generalization ability.
>
> Our patch-embedding layer provides a degree of generalizability due to its modality-specific design. Furthermore, it can effectively accommodate new modalities.
> * We currently use three types of modality-specific patch-embedding layers that can handle 10 tasks. If a new task fits within these three modalities, there's no need to add a new embedding layer. For instance, the carbon dioxide image translation task using a new dataset can directly utilize the existing embedding layer without requiring any modifications.
> * If a new modality emerges, our framework can adapt by adding a new modality-specific patch-embedding layer to accommodate tasks related to this new modality. As this is the first work on a weather generalist foundation model, there is potential for future advancements to enhance our model's adaptability across different modalities.
>
> #### W2: Comparison with other climate foundation models
>
> >It would be beneficial to compare WeatherGFM’s performance against other climate foundation models (e.g., Aurora, ClimaX) using the ERA5 dataset.
>
> To address your concerns, we have expanded our tasks to include atmospheric variables forecasting. We compared our WeatherGFM with Climax and the operational Integrated Forecasting System (IFS) on WeatherBench. Table 1 shows that our approach significantly outperforms ClimaX in the 120-hour and 168-hour forecast results, even surpassing the IFS method of ECMWF. Our method achieves comparable results to ClimaX at other lead times. It is important to note that Climax fine-tunes the pre-trained model for each lead time, necessitating N models for weather forecasting at N lead times. In contrast, our generalist model can manage all lead time tasks with a single model, eliminating the need for task-specific fine-tuning.
>
> **Note**: Comparison of RMSE and ACC across different lead times for IFS, ClimaX, and ours WeatherGFM on the temperature at 2 meters above ground (T2m) variable prediction task. ClimaX views predicting at each lead time as a separate task and *fine-tunes a separate model for every individual task*. In contrast, **our WeatherGFM utilizes a single model to handle all of these tasks**.
> | Lead Time(hr.) | RMSE$\downarrow$ | RMSE$\downarrow$ | RMSE$\downarrow$ | ACC$\uparrow$ | ACC$\uparrow$ | ACC$\uparrow$ |
> |-----------|------|--------|------|------|--------|------|
> |           | IFS  | ClimaX | WeatherGFM | IFS  | ClimaX | WeatherGFM |
> | 6         | **0.97** | 1.11   | 1.08 | **0.99** | 0.98   | 0.98 |
> | 24        | **1.02** | 1.19   | 1.23 | **0.99** | 0.97   | 0.97 |
> | 72        | **1.30** | 1.47   | 1.56| **0.98** | 0.96   | 0.96 |
> | 120       | 1.71 | 1.83   | **1.68** | **0.96** | 0.94   | 0.95 |
> | 168       | 2.23 | 2.17   | **1.76** | 0.93 | 0.91   | **0.94** |
>
> #### W3: Unclear Sources of Performance Gains
> >It’s unclear whether the performance gains are due to dataset scalability or multi-task training or larger number of parameters.
>
> In our paper, Figure 7 demonstrates the improvements in model scale and parameters across various tasks. As the number of tasks increases, the data size also expands. We have combined these two aspects to assess performance enhancements derived from both data and tasks.
>
> As shown in the table below, WeatherGFM-4tasks is trained on four tasks: Radar Temporal SR, GOES2GOES, GOES2Radar, and Radar Spatial SR. It uses more data and encompasses more tasks than the ViT-ST specialized model, which is trained on these four tasks separately. However, WeatherGFM-4tasks utilizes less data and fewer tasks compared to WeatherGFM-10tasks, which is trained on ten tasks.
>
> The results indicate that for radar image generation tasks (Radar Temporal SR, GOES2Radar, and Radar Spatial SR), both WeatherGFM-4tasks and WeatherGFM-10tasks outperform ViT-ST. This is likely because most of the selected tasks are related to radar image generation. However, for the satellite image generation task (GOES2GOES), WeatherGFM-4tasks does not perform as well. This suggests that multi-task learning of similar tasks can enhance the model's performance on those tasks.

---

> ### Author Response · Authors · 2024-11-23
> **Response to Reviewer vbFm (Part 2)**
>
> ##### The Effect of Dataset Size
>
> *Note: ViT-ST: single-task ViT. WeatherGFM-4Tasks: our WeatherGFM trained on 4 tasks. WeatherGFM-10Tasks: our WeatherGFM trained on full (10) tasks.*
>
>
> ##### Radar Temporal SR
>
> | Metrics   | RMSE$\downarrow$|CSI/74$\uparrow$ | CSI/133$\uparrow$ | CSI/160$\uparrow$ | CSI/181$\uparrow$ | CSI/219$\uparrow$ |
> |-----------|----|----|---------|---------|---------|---------|
> | ViT-ST  | 0.333 | 0.585  | 0.366   | 0.282   | 0.215   | 0.063   |
> | WeatherGFM-4Tasks | 0.353 | 0.576  | 0.355   | 0.274   | 0.209   | **0.074**   |
> | WeatherGFM-10Tasks | **0.327** | **0.597**  | **0.376**   | **0.287**   | **0.217**   | 0.073   |
>
> ##### GOES2GOES
>
> | Metrics |RMSE$\downarrow$  | CSI/-6K$\uparrow$ | CSI/-4K$\uparrow$ | CSI/0$\uparrow$  | CSI/2K$\uparrow$ |
> |--------|---|---------|---------|--------|--------|
> | ViT-ST | **0.257** | 0.987   | **0.972**   | **0.809**  | 0.136  |
> | WeatherGFM-4Tasks | 0.317 | **0.994**   | 0.968   | 0.766  | 0.148  |
> | WeatherGFM-10Tasks | 0.310 | 0.993   | 0.968   | 0.808  | **0.222**  |
>
> ##### GOES2Radar
>
> | Metrics |RMSE$\downarrow$  | CSI/74$\uparrow$ | CSI/133$\uparrow$ | CSI/160$\uparrow$ | CSI/181$\uparrow$ | CSI/219$\uparrow$ |
> |--------|---|--------|---------|---------|---------|---------|
> | ViT-ST  |  0.445| 0.424  | 0.242   | 0.181   | 0.134   | 0.045   |
> | WeatherGFM-4Tasks |0.460 | 0.443  | 0.263   | **0.213**   | **0.166**   | **0.059**   |
> | WeatherGFM-10Tasks |**0.436** | **0.447**  | **0.266**   | 0.208   | 0.157   | 0.053   |
>
> ##### Radar Spatial SR
>
> | Metrics | RMSE$\downarrow$ | CSI/74$\uparrow$ | CSI/133$\uparrow$ | CSI/160$\uparrow$ | CSI/181$\uparrow$ | CSI/219$\uparrow$ |
> |---------|--|--------|---------|---------|---------|---------|
> | ViT-ST  | **0.120** | 0.820  | 0.703   | 0.634   | 0.573   | 0.387   |
> | WeatherGFM-4Tasks | **0.120** | **0.831**  | **0.714**   | **0.646**   | **0.574**   | **0.380**   |
> | WeatherGFM-10Tasks | 0.121 | **0.831**  | 0.712   | 0.644   | 0.570   | 0.375   |
>
>
>
> #### W4: Weaker performance in weather image translation tasks
>
> > Why does WeatherGFM show weaker performance in weather image translation compared to other tasks?
>
> * As demonstrated in the table below, after an additional 5 epochs of training, we observed an improvement in the performance of weather image translation (GOES-IR2GOES-Visible task). We have updated the corresponding results, which are highlighted in blue in Table 2 of the main paper.
>
> ##### GOES-IR2GOES-Visible
>
> | Metrics |RMSE$\downarrow$  | CSI/2000$\uparrow$ | CSI/3200$\uparrow$ | CSI/4400$\uparrow$ | CSI/5600$\uparrow$ | CSI/6800$\uparrow$ |
> |--------|---|--------|---------|---------|---------|---------|
> | UNet  |  0.915 | 0.422  | 0.285  |  0.179   | 0.100   | 0.040   |
> | ViT |0.448 | 0.574  | 0.437   | **0.303**   | **0.184**   | **0.071**  |
> | WeatherGFM |**0.439** | **0.580**  | **0.439**  | 0.298   | 0.166   | 0.068   |
>
> * The primary reason for the weaker performance in this task is that only the GOES-IR2GOES-Visible task uses GOES-Visible as its generation target. In contrast, when radar data is the generation target, it encompasses a greater number of tasks, such as Radar Extrapolation and GOES2Radar. As a result, the GOES-IR2GOES-Visible task is more challenging to train and requires more training epochs to achieve improved results.
> * In fact, different tasks have varying levels of learning difficulty. Some simple tasks are difficult to be improved further since they can be easily trained to a relatively high performance level. This phenomenon can also be witnessed in PromptGIP [1] and GenLV [2]. In our experiments, the satellite radar image super-resolution task is simpler compared to other tasks. For example, the CSI index for the satellite super-resolution task has reached around 0.99. Our model has achieved sufficiently good performance.
>
> [1] Liu Y, Chen X, Ma X, et al. Unifying image processing as visual prompting question answering[J]. arXiv preprint arXiv:2310.10513, 2023.
>
> [2] Chen X, Liu Y, Pu Y, et al. Learning a low-level vision generalist via visual task prompt[C]//Proceedings of the 32nd ACM International Conference on Multimedia. 2024: 2671-2680.
>
>
> #### W5: A misuse of double lines for "GOES2Radar, CSI/219."
>
> Thank you for your suggestion. We have made the revisions.

---

> ### Author Response · Authors · 2024-11-27
> **Kindly Requesting for Review of Our Response**
>
> Dear Reviewer vbFm,
>
> **We are eager to ensure that we have adequately addressed your concerns** and are prepared to offer further clarifications or address any additional questions you may have.
>
> Should you find that our revisions have satisfactorily addressed your previous concerns, **we would be most grateful if you would reconsider the evaluation of our paper with a view to enhancing its standing**.
>
> We would like to express our heartfelt gratitude for the time and effort you have dedicated to reviewing our work. **It has been a pleasure to engage with you throughout this process**.
>
>
> Best regards,
>
> Authors

---

> ### Author Response · Authors · 2024-11-28
> **Gentle Reminder**
>
> Dear Reviewer vbFm,
>
> We hope this message finds you well. As we are nearing the end of rebuttal discussion, we would greatly appreciate it if you could find some time in your busy schedule to review our response and let us know if you have any further questions or require additional clarification on any points.
>
> We sincerely appreciate your valuable time and effort.
>
> Best Regards,
>
> The Authors

---

> ### Author Response · Authors · 2024-11-30
> **Kind Reminder**
>
> Dear Reviewer vbFm,
>
> We hope this message finds you well. We apologize for any inconvenience caused by reaching out over the weekend. We noticed that your initial rating of our work was slightly negative. As the rebuttal discussion period is coming to a close, we would greatly appreciate your feedback on whether our rebuttal has effectively addressed your concerns. Please let us know if you have any further questions, and we would be more than happy to assist you.
>
> We sincerely appreciate your valuable time and effort.
>
> Best Regards,
>
> The Authors

---

> ### Author Response · Authors · 2024-12-01
> **Kind Reminder**
>
> Dear Reviewer vbFm,
>
> We hope this message finds you well. We apologize for reaching out again over the weekend. As **the rebuttal discussion period ends in two days**, we would greatly appreciate your review of our rebuttal. We are eager to ensure we have fully addressed your concerns and would be happy to answer any further questions you may have.
>
> We sincerely appreciate your valuable time and effort.
>
> Best Regards,
>
> The Authors

---

> ### Author Response · Authors · 2024-12-02
> **Gentle Reminder: Rebuttal Discussion Deadline Approaching**
>
> Dear Reviewer vbFm,
>
> We hope this message finds you well. We are writing to gently remind you that the rebuttal discussion period will conclude in less than 30 hours. We are keen to ensure that we have thoroughly addressed all your concerns and are ready to respond to any additional questions or clarifications you might have.
>
> Your insights are extremely valuable to us, and we greatly appreciate the time and effort you have dedicated to reviewing our work. Please let us know if there is anything further we can provide to assist in your review.
>
> Thank you once again for your attention and support.
>
> Best regards,
>
> The Authors

---

> ### Author Response · Authors · 2024-12-03
> **Kind Reminder**
>
> Dear Reviewer vbFm,
>
> We hope this message finds you well. We apologize for reaching out again, but with the rebuttal discussion period closing in the next 5 hours, we wanted to draw your attention to some key points in our response.
>
> In particular, we would like to highlight the additional experiments conducted on the ERA5 dataset. Our model has outperformed other fundation models (e.g., ClimaX) in some representative variables. Furthermore, we have included experiments with varying data sizes, which demonstrate that multi-task learning of similar tasks can enhance the model's performance.
> We sincerely hope that our clarifications might encourage a reevaluation of our work.
>
> We appreciate your understanding and your time. Please let us know if there is anything further we can provide to assist in your review.
>
> Best regards,
>
> The Authors

---

> > ### Comment · Reviewer_vbFm · 2024-12-03
> >
> > First, I want to thank the authors for responding to my comments, and I apologize for the delay in my response.
> >
> > My main concerns were the task-specific embedding layer’s generalization to unseen tasks, the performance of WeatherGFM with varying dataset sizes, and comparisons with other foundation models such as Aurora and Climax. After reviewing the authors’ responses and discussions with other reviewers, I feel that most of my concerns have been addressed. As a result, I will raise my score from 5 to 6.
> >
> > I would like to elaborate further on WeatherGFM's performance and the reasoning behind my decision. In my evaluation, I carefully reviewed the discussion between Reviewer CC3i and Reviewer qe7B. Reviewer CC3i highlighted that WeatherGFM shows lower performance than single-task ViT on certain tasks. I believe that scaling WeatherGFM with larger datasets could potentially address this issue. However, given the inherent challenges of collecting weather datasets, I find it unreasonable to expect the authors to gather significantly larger datasets, and thus I do not think it is fair to make this request.
> >
> > Additionally, I agree with Reviewer qe7B's perspective that this work is an important step toward developing the first generalist model in the weather domain. From this standpoint, the comparisons between WeatherGFM, UNet, and ViT are appropriate and meaningful. However, the lower performance on specific tasks compared to simple single-task baselines still remains as a limitation. I appreciate the authors sharing their thoughts on these tasks in Response Part 2, and I believe that incorporating the discussion of potential solutions into the main paper would further strengthen this work.

---

> > > ### Author Response · Authors · 2024-12-03
> > > **Thank you**
> > >
> > > Dear Reviewer vbFm,
> > >
> > > We sincerely appreciate your response and are pleased to know that we have largely addressed your concerns. We are truly grateful for your understanding and recognition of our efforts toward developing the first generalist model in the weather domain. Your forward-looking perspective on the field is encouraging, and we will ensure that the revised version of our main paper includes a discussion of potential solutions in response to your feedback.
> > >
> > > Once again, we appreciate your revised rating, your valuable input, and the time you have invested in reviewing our paper. Your feedback has significantly contributed to enhancing the quality of our work.
> > >
> > > Thank you once again for your valuable feedback.
> > >
> > > Best regards,
> > >
> > > The Authors

---

### Author Response · Authors · 2024-12-02
**General Comments and Revision Summary**

We sincerely thank all the reviewers for their insightful reviews and helpful suggestions, as well as for acknowledging the importance, comprehensiveness, and contributions of our work. We have submitted a revised version of our paper and provide a summary of the revisions below:

1. Based on the suggestions from reviewers vbFm and CC3i, we have added extended experiments on the ERA5 dataset to demonstrate our framework's extensibility. We compared our model's performance on two commonly used variables (T2m, U10) with other climate foundation models (e.g., ClimaX) and demonstrated better results.

2. Following reviewer vbFm's feedback, we have also included experiments with varying data sizes. The results indicate that multi-task learning of similar tasks can enhance the model's performance.

3. Reviewers Broi and qe7B requested more details on the prompt design and the mixed-modal masked image modeling (MMIM) pipeline. We have clarified these details and included them in the paper.

4. All reviewers noted WeatherGFM's weaker performance in some tasks. It is important to recognize that different tasks have varying levels of learning difficulty. Some simpler tasks are challenging to improve further because they are already learned to a high performance level. This phenomenon is also observed in existing works. The most critical advantages of our method over specialized models are its versatility and generalization ability. While specialized, data-driven models struggle to generalize to new tasks, our model demonstrates versatility across multiple tasks and generalization ability on unseen tasks without any fine-tuning.

Lastly, we would like to express our gratitude once again to all the reviewers for their constructive feedback, which has greatly improved our paper. We also hope that the reviewers will be available for further review discussions.

---

### Meta-Review · Area_Chair_uZAQ · 2024-12-20

**Metareview:**

This paper introduces a generalist model for various weather understanding tasks. The key contribution is demonstrating that the widely used principles for vision generalist models—framing diverse image-to-image translation tasks as a unified in-context learning problem—can be effectively extended to multi-modal weather data. The authors validate their approach across a range of weather tasks involving diverse modalities, showcasing its potential versatility.

The paper received mixed reviews, including one strong positive, two borderline positive, and one strong negative assessment. The primary concerns raised by the reviewers were as follows:

1. Missing comparisons to other generalist models(**vbFm**, **CC3i**),
2. Limited technical novelty (**CC3i**, **qe7B**),
3. Constrained generalization to other modalities (**vbFm**, **CC3i**), and
4. Choice of weak baselines (**vbFm**, **CC3i**).

In response, the authors provided significant clarifications and new results, including comparisons to foundation models (e.g., ClimaX), analysis of scale-performance correlation, and quantitative evaluations on out-of-distribution (OOD) tasks.

The AC thoroughly reviewed the paper, reviews, and rebuttal discussions. The AC concludes that the paper is on the borderline. On the critical side, the paper naively adapts the framework of PromptGIP (Liu et al., 2024) to weather data, with the primary extension being modality-specific input embedding layers. While a defining characteristic of foundation models is their ability to generalize to downstream tasks beyond the pre-training data (e.g., as seen in ClimaX and Aurora), the presented experiments mostly emphasize multi-task learning within the pre-training data, failing to stretch further from PromptGIP. However, the preliminary results on the WeatherBench dataset, provided during the rebuttal, are promising and partially demonstrate the potential of the proposed method for OOD generalization, even though the evidence remains limited.

Considering that the paper addresses an important problem and the rebuttal results on WeatherBench show encouraging signs of OOD generalization, the AC recommends **acceptance**. To strengthen the work, the authors are encouraged to:

1. Supplement the preliminary comparison results with additional analyses against other generalist models.
2. Clearly present experimental details to improve the credibility of the findings (e.g., clarifying fine-tuning datasets for ClimaX and WeatherGFM, and including more results).
3. Release the code to facilitate reproducibility and further exploration by the research community.

**Additional Comments On Reviewer Discussion:**

Several reviewers raised significant concerns about the paper. These include limited technical novelty, insufficient experiments comparing the proposed model to other foundation models and stronger task-specific baselines, inadequate validation of out-of-distribution (OOD) generalization capabilities, and unclear presentation. In response, the authors provided extensive clarifications and additional results, such as comparisons to foundation models (e.g., ClimaX), an analysis of scale-performance correlation, and quantitative evaluations on OOD tasks. Reviewers vbFm and Broi found the responses satisfactory and raised their scores to weak accept. However, reviewer CC3i remained unconvinced, citing that the rebuttal results were still far from comprehensive compared to other weather foundation models. Reviewer qe7B strongly supported the paper's acceptance, but the review may have been slightly too optimistic. The AC agrees with reviewer CC3i's concerns but leans slightly towards acceptance since the preliminary results look promising.

---

### Decision · Program_Chairs · 2025-01-22

Accept (Poster)